


# Predicting atmospheric background number concentration of ice nucleating particles in the Arctic

Guangyu Li[1,*], Jörg Wieder[1,*], Julie T. Pasquier[1], Jan Henneberger[1], and Zamin A. Kanji[1]

[1]Institute for Atmospheric and Climate Science, ETH Zurich, Switzerland
[*]These authors contributed equally to this work.

**Correspondence:** Guangyu Li (guangyu.li@env.ethz.ch), Jörg Wieder (joerg.wieder@env.ethz.ch), Zamin A. Kanji (zamin.kanji@env.ethz.ch)

**Abstract.** Mixed-phase clouds (MPCs) can have a net warming or cooling radiative effect on climate influenced by the phase and concentration of cloud particles. They have received considerable attention due to high spatial coverage and occurrence frequency in the Arctic. To initiate ice formation in MPCs at temperatures above -38 °C, ice nucleating particles (INPs) are required, which therefore have important implications on the radiative properties of MPCs by altering the ice to liquid ratio of hydrometeors. As a result, constraining ambient INP concentrations could promote accurate representation of cloud microphysical processes and reduce the uncertainties in estimating the cloud-phase-related climate feedback in climate models. Currently, INP parameterizations are lacking for remote Arctic environments. Here we present INP number concentrations and their variability measured in Ny-Ålesund (Svalbard) at temperatures between 0 and -30 °C. No distinguishable seasonal difference was observed from 12 weeks of field measurements in autumn 2019 and spring 2020. In addition, correlating INP concentrations to aerosol physical properties was not feasible. Therefore, we propose a lognormal-distribution-based parameterization to predict Arctic INP concentration solely as a function of temperature. In practice, the parameterized variables allow for a) the prediction of the most likely INP concentrations and; b) the retrieval of the governing distribution of INP concentrations at given temperatures in the Arctic.

## 1 Introduction

The Arctic region is extremely sensitive to climate change. During the past several decades, this region has undergone accelerated warming more than twice the rate of the global average (Serreze and Barry, 2011) - a phenomenon termed Arctic amplification. Many feedback mechanisms are considered to contribute to the rapid warming of the Arctic environment. For instance, the phase partitioning in MPCs, i.e., the ratio of supercooled liquid droplets and ice crystals, markedly determines the cloud optical depth and therefore impacts the radiative budget of the Arctic boundary layer (Pithan and Mauritsen, 2014).

In Arctic MPCs, primary ice formation is facilitated via heterogeneous ice nucleation aided by INPs (Vali et al., 2015). Despite the scarcity of approximately only 1 out of $10^6$ total aerosol particles acting as INP (at -20 °C) in the free troposphere (DeMott et al., 2010; Kanji et al., 2017), the variation of INP concentration can indirectly affect the climate by modifying cloud microphysical and optical properties and producing precipitation (Lohmann, 2002; Mason et al., 2015). Therefore, global and regional climate models require accurate representations of complex cloud microphysical processes and INP feedback. So far,





knowledge gaps still exist concerning the spatial and seasonal variations, chemical compositions, and source origins of INPs, particularly in remote Arctic regions (Hartmann et al., 2020).

Mineral dust and soil particles are effective INPs at temperatures lower than approximately -15 °C (Hoose and Möhler, 2012; Murray et al., 2012; Kanji et al., 2017), yet the number is proportionally low in the Arctic due to reduced sources. In addition, recent studies (Wilson et al., 2015; DeMott et al., 2016; Irish et al., 2017; McCluskey et al., 2018; Twohy et al., 2021) present

evidence that the emission of sea spray aerosol (SSA) via the bubble bursting mechanism at the ocean surface (Gantt and Meskhidze, 2013) can be a dominant INP source in remote regions, e.g., the southern ocean, where other active INPs sources (e.g., mineral dust) are rare. Many recent studies attempted to quantify INP concentrations in diverse environments and develop deterministic parameterizations to represent cloud microphysical processes in climate models. For instance, DeMott et al. (2010) (referred to as D10) incorporate the global average INP observations and improved INP parameterization by relating

the dependence of INP concentrations on temperature and number concentrations of aerosol particles with diameters above 0.5 μm. Tobo et al. (2013) and Schneider et al. (2021) developed INP parameterizations correlated with fluorescent biological particles and ambient temperatures, respectively, from measurement campaigns in pine forest ecosystems where biological aerosols dominate the INP population. In addition, many surface site density ($n_s$) based INP parameterization studies focused on prevailing INP sources in different environments, e.g., mineral dust (Niemand et al., 2012 (N12); DeMott et al., 2015 (D15)),

and pristine SSA (McCluskey et al., 2018, M18). The dominant aerosol compositions define the major differences in these INP parameterizations, i.e., the slope of INP number concentrations as a function of temperature. Due to the strong temperature dependence, the slope of an INP parameterization was reported to alter the amount of outgoing radiation by reforming the vertical distribution of cloud microphysical processes in modeling studies (Hawker et al., 2021). However, the community still lacks an INP parameterization capable of predicting the INP number concentrations in pristine regions such as the Arctic. In

particular, the previously mentioned parameterizations are not suitable for remote pristine conditions (see Section 3.2).

Apart from the contribution of localized INP sources, remote effects cannot be ruled out (Schmale et al., 2021) such that the changing aerosol emissions at low and mid-latitudes will also impact the Arctic region (Najafi et al., 2015; Lewinschal et al., 2019). Igel et al. (2017) suggested that long-range transport can be an increasing aerosol source within the framework of climate change and can impact the low-level Arctic cloud cover where the local aerosol loading is less prevalent. Additionally,

Schrod et al. (2020) revealed that the dominant INP species vary temporally and geographically, which complicates the realistic representativeness of atmospheric INPs in the Arctic. As a result, the Arctic INP population is considered to have a well-mixed composition from marine and terrestrial origin (Murray et al., 2021), and remote and local effects cannot be easily distinguished (Schmale et al., 2021). Ott (1990) suggested that in many atmospheric processes, a substance of interest, e.g., aerosol or INP, undergoes random dilution and mixing during atmospheric transport. The resulting frequency of INP concentrations at the

destination, e.g., the Arctic, in this case, converges to a log-normal distribution after successive random dilutions in the absence of dominating local sources.

INP-related measurements in the remote Arctic are scarce and therefore of extreme value where the ambient aerosols are in well-mixed and near-pristine conditions. In this study, we conducted continuous measurements over 12 weeks in the Arctic gathering extensive INP-related measurement data set, a broader INP-temperature spectrum down to -30 °C, and higher tem-



poral resolutions. By quantifying the distribution of INP concentrations, we developed an INP parameterization representative of atmospheric background air masses. This parameterization will help evaluate the role of cloud phase interactions in Arctic MPCs, and contribute to the progress on accurately estimating cloud influenced climate predictions in the Arctic (Tan and Storelvmo, 2019).

## 2  Methods

### 2.1  Overview of field campaign and experimental setup

Under the framework of the NASCENT campaign (Pasquier et al., 2021, in revision, BAMS) ambient INP measurements and aerosol characterization took place at the Arctic field site in Ny-Ålesund, Svalbard (78.9 °N, 11.9 °E) during October–November 2019 and March–April 2020. Ny-Ålesund is located on the south coast of the Kongsfjorden in western Svalbard, and it is a well-established international site in the Arctic for scientific research. We configured ambient INP and aerosol
measurements in a container used as the temporary observatory, which was placed at the southern end of Ny-Ålesund town (see Fig. 1a). In addition to the minor anthropogenic emissions from the town, the container was approximately 600 m from the coast of Kongsfjorden, and was surrounded by mountains and glaciers.

The experimental setup is shown in Fig. 1b. Ambient aerosol was sampled through a custom-built total aerosol inlet of $4.5$ m vertical length, which was kept at a maximal temperature of $40$ °C. During very cold periods with strong winds (e.g.
ambient temperature of -30 °C with wind chill below -50 °C in March 2020) the temperature temporarily dropped, however, never below 0 °C. A flow splitter (custom-built), a $0.5$ m long 90°-bend and a three-way ball valve (Model 120VKD025-L, Pfeiffer Vacuum, Germany) connected a blower (Model U71HL, Micronel AG, Switzerland) and a high flow-rate impinger (Coriolis® µ, Bertin Instruments, France), both operating at $300$ L min$^{-1}$, downstream to the inlet (similarly to the setup used in Wieder et al. (2021) at Wolfgangpass in Davos). The blower was used to create an offset flow through the inlet during times
the impinger was not collecting aerosol for INP analysis. In addition, the total ambient flow was divided into a home-built online continuous flow diffusion chamber - HINC (Lacher et al., 2017), which sampled ambient aerosol continuously at a flow rate of $0.283$ std L min$^{-1}$ (Details in Fig. 2).

### 2.1.1  INP sampling and measurements

Using the Coriolis impinger with the cut-off size of $0.5$ µm (aerodynamic diameter), ambient aerosol samples were collected
into pure water and consecutively analyzed for INP concentration on site using the offline drop freezing technique DRINCZ (David et al., 2019). To compensate the evaporation loss of the sampling liquid (W4502-1L, Sigma-Aldrich, US) during the operation of the impinger, additional sampling liquid was fed into the sampling cone at a constant feed rate, which varied according to the ambient conditions and ranged between $0.6$ and $1.0$ mL min$^{-1}$. Between October 6 and November 15 in 2019, and from March 16 to April 22 in 2020, 137 and 133 samples, respectively, were analyzed for INP concentration. The





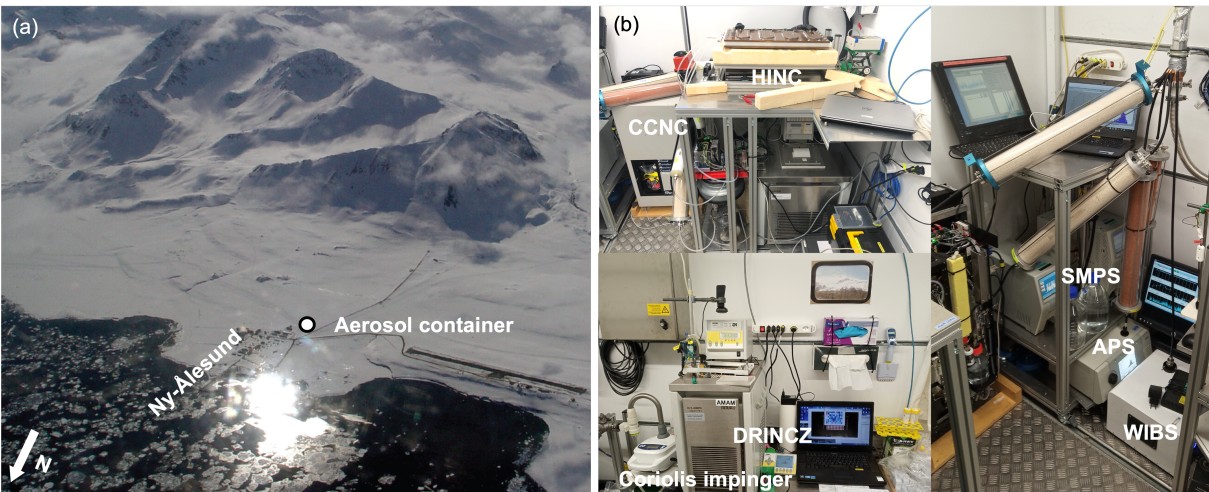

**Figure 1.** On-site instrumental setup in Ny-Ålesund. (a) Location of container for ambient INP and aerosol measurement (Photo CC BY Radovan Krejci). (b) In-container setup. Ambient aerosol flow was directed into CCNC (Cloud Condensation Nuclei Counter), HINC (Horizontal Ice Nucleation Chamber, Lacher et al., 2017), Coriolis impinger and DRINCZ (DRoplet Ice Nuclei Counter Zurich, David et al., 2019), SMPS (Scanning Mobility Particle Sizer) and APS (Aerodynamic Particle Sizer), WIBS (Wideband Integrated Bioaerosol Sensor).

INP concentration was calculated at every integer temperature according to Vali (1971, 2019) as

$$N_{\text{INP}}(T) = -\frac{\ln\left[1 - \frac{N_{\text{fro}}(T)}{N_{\text{tot}}}\right]}{V_{\text{a}} \cdot C} \tag{1}$$

where $N_{\text{fro}}(T)$ is the number of frozen aliquots at temperature $T$, $N_{\text{tot}}$ is the total number of aliquots ($N_{\text{tot}} = 96$), and $V_{\text{a}}$ is the volume of an individual aliquot ($V_{\text{a}} = 50\,\mu\text{L}$). $C$ is the normalization factor in order to calculate the INP concentration per standard liter of sampled air (std L$^{-1}$) and is defined as

$$C = \frac{F_{\text{Coriolis}} \cdot t_{\text{sample}}}{V_{\text{Coriolis}}} \cdot \frac{p_{\text{ambient}}}{p_{\text{std}}} \cdot \frac{T_{\text{std}}}{T_{\text{ambient}}} \tag{2}$$

where $F_{\text{Coriolis}}$ is the flow rate of the impinger (300 L min$^{-1}$), $t_{\text{sample}}$ is the sampling time (60 minutes), $V_{\text{Coriolis}}$ is the end volume within the sampling cone (15 mL), $p_{\text{ambient}}$ and $p_{\text{std}}$ (1013.25 hPa) are the ambient and standard-condition pressure, and $T_{\text{ambient}}$ and $T_{\text{std}}$ (273.15 K) are the ambient and standard-condition temperature. The retrieved INP concentrations were corrected for the background of blank samples according to David et al. (2019). From the analysis by DRINCZ, the highest

temperature at which the INP concentration could be measured was approximately -5 °C, while the coldest temperature where ice nucleation activity was reliably observed is -22 °C.

In order to extend the INP temperature spectrum, we utilized the online continuous flow diffusion chamber HINC (Lacher et al., 2017) to measure INP concentrations at $T = -30$ °C ($\pm0.4$°C uncertainty) at a relative humidity with respect to water $RH_{\text{w}} = 104$ % ($\pm1.5$ % uncertainty), representing the immersion/condensation freezing mode. The ice crystals and water





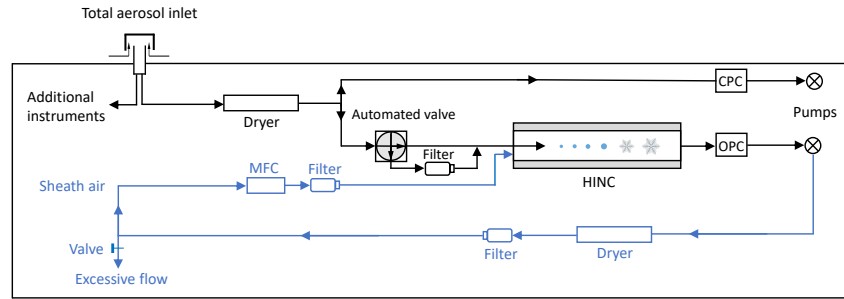

**Figure 2.** Flow diagram of HINC as setup inside the container during field measurements in Ny-Ålesund (Figure adapted from Lacher et al., 2017). The blue part indicates modifications from Lacher et al. (2017) to adapt the recirculation of sheath air. MFC, CPC and OPC represent mass flow controller, condensation particle counter and optical particle counter, respectively.

droplets were distinguished by a pre-determined size threshold (5 µm) of an optical particle counter (OPC) downstream of the chamber (see Fig. 2), considering the operation temperature and particle residence time. To account for the false-positive ice count originated from the internal chamber, e.g., falling frost from the warmer plate, a motorized valve (see Fig. 2) was applied to switch from sample flow to filtered air measurements regularly (5 min) before and after each sampling period (15 min) to determine a background counts, which were used to determine the limit of detection (LOD) of the instrument at the

measuring conditions (details in Lacher et al., 2017). Overall, 348 and 594 15-min samples were collected using HINC during the campaign in 2019 and 2020, respectively. At $T = -30$ °C, only measured INP concentration larger than the LOD are presented and used in developing the parameterization, given the 68.3 % confidence interval of significance according to the Poisson statistics. For Autumn 2019 and Spring 2020, the number of reported INP observations above the LOD were 135 and 323, with the median INP concentrations of 2.16 and 3.16 std L$^{-1}$, and LOD equal 1.06 and 1.12 std L$^{-1}$, respectively.

### 2.1.2 Aerosol physical property measurements

Particle size distributions were recorded by two commercial particle sizer spectrometers connected to the total inlet splitter. Coarse particles (approximately 0.5 - 20 µm) were detected using an Aerodynamic Particle Sizer Spectrometer (APS, Model 3321, TSI Corp., US). Fine particles (Aitken and accumulation mode, range approximately from 15 - 600 nm) were detected using a Scanning Mobility Particle Sizer Spectrometer (SMPS, Model 3938, TSI Corp., US). Electrical mobility diameters of

the SMPS and aerodynamic diameters of the APS were converted to volume equivalent physical diameter assuming an average particle density of 2.0 g cm$^{-3}$ (Tobo et al., 2019) and a shape factor of 1.2 (Thomas and Charvet, 2017). The concentration of fluorescent particles was observed using a Wideband Integrated Bioaerosol Sensor (WIBS-5/NEO, DMT, US).



## 2.2 Parameterization approach

The observed INP concentrations were obtained by two instruments, with measurements at $T = -30\,°C$ using HINC and from
$T = -22$ to $-5\,°C$ using DRINCZ. Consider a data set with $n$ observations and $p$ variables (i.e., different measured tem-
peratures). The relationship between $Y \in \mathbb{R}^{n \times p}$ (i.e., INP concentrations in logarithmic scale) and $X \in \mathbb{R}^{n \times p}$ (i.e., measured
temperatures) can be fit with a linear regression with slope vector $\beta$, including the errors of the observations $\epsilon$:

$$Y = X\beta + \epsilon, \tag{3}$$

The linear regression with ordinary least squares (OLS) assumes constant variance in the errors (i.e., homoscedasticity). By
applying OLS linear regression, however, as seen in Fig. A1 in the residual plot, the heteroscedastic INP concentrations over
each measured temperature are observed, which motivates the use of weighted least squares (WLS, see e.g. Strutz (2010))
linear regression to scale the median log-normal fit of INP concentrations at different temperatures. In WLS, the error term $\epsilon$
is assumed to be normally distributed with the mean value of 0 and non-uniform variance-covariance matrix $E$:

$$E = \begin{pmatrix} \sigma_1^2 & 0 & \cdots & 0 \\ 0 & \sigma_2^2 & \cdots & 0 \\ \vdots & \vdots & \ddots & \vdots \\ 0 & 0 & \cdots & \sigma_n^2 \end{pmatrix} \tag{4}$$


We take the heteroscedasticity into account by dividing each observation by assigning extra non-negative weights $w_i$. Let
the matrix $W$ be a diagonal matrix containing these weights:

$$W = \begin{pmatrix} w_1 & 0 & \cdots & 0 \\ 0 & w_2 & \cdots & 0 \\ \vdots & \vdots & \ddots & \vdots \\ 0 & 0 & \cdots & w_n \end{pmatrix} \tag{5}$$

The weighted least squares estimate is then:

$$\hat{\beta}_{\text{WLS}} = \arg\min_{\beta} \sum_{i=1}^{n} \epsilon_i^2 = (X^T W X)^{-1} (X^T W Y), \tag{6}$$

To minimize the effect of uneven distribution of the observed data set, the weight size in the weighing matrix $W$ needs to
be properly determined. In WLS, a typical weighing factor is to scale the standard deviation of the error $\epsilon_i$ (i.e., $w_i = 1/\sigma_i$).
However, the strength and other factors depending on the intrinsic properties of the data could play a role when determining





weighing sizes. As a result, to develop a linear regression within our observed INP concentrations in the exponential area and temperatures, we selected three different weighing matrices $W$, i.e., WLS_$W_\sigma$, WLS_$W_{obs}$ and WLS_$W_{\sigma^2}$, representing weighing factor of $1/\sigma_i$, number of observations at each measured temperature, and $1/\sigma_i^2$, respectively. In Table A1, we compared the fitting parameters generated by applying different linear regression methods to the observation data set. The highest r$^2$ and the lowest RMSE and MAPE values were obtained for the WLS method when weighing the uneven distribution

by the number of observations at each measured temperature. Therefore, the number of observations at each temperature was chosen as the weighing factor in WLS for developing the parameterization for INP concentrations. The authors suggest using the weighted least square method for future field studies when intercomparing the measurements from diverse instruments and measuring conditions to reduce the systematic bias.

## 3 Results and discussion

### 3.1 Overview of Arctic ambient INP concentrations

An overview of the observed INP number concentrations as a function of temperature at Ny-Ålesund (Svalbard, Norway) within the framework of the Ny-Ålesund AeroSol Cloud ExperimeNT (NASCENT) campaign (Pasquier et al., 2021, in revision, BAMS) for autumn 2019 and spring 2020 is shown in Fig. 3. The overview of the measurement site and the experimental setup are given in the Section 2.1 and Fig. 1a. INP concentrations at -30 °C were measured using the Horizontal Ice Nucleation

Chamber (HINC, Lacher et al., 2017). INP spectra above -22 °C were measured with the DRoplet Ice Nuclei Counter Zurich (DRINCZ, David et al., 2019) using the liquid samples collected by a high flow-rate impinger (Further details given in Section 2.1.1). The INP measurements at T < -22 °C were lacking because all droplets froze in most cases, hindering the calculation of INP concentrations. While concerning observations at T > -5 °C, most INP concentrations were below the limit of detection (LOD) of the instrument due to the volume of air sampled, and thus could not be reliably derived. Overall, in Fig. 3, the

INP concentrations were found to increase in a virtually log-linear pattern with the decreasing temperature for both observing periods (orange line). To detect the seasonality of the observed INP concentration, we conducted unpaired $t$-tests with a significance level of 5 % for observed INP concentrations at each temperature. Overall seasonal variation of INP concentrations at all measured temperatures is indistinguishable.

### 3.2 Relationship between INP concentrations and aerosol properties

State-of-the-art parameterizations predict INP concentration based on an aerosol property such as number or surface concentration (e.g., D10, N12, D15, M18). We investigated the effectiveness of aerosol properties as predictors for INP concentration at different temperatures. Figure 4 shows the observed INP concentrations for both seasons as a function of the concentration of particles with a diameter larger than 0.5 μm ($n_{\geq 0.5 \mu m}$). Similarly, other aerosol properties, i.e., total surface area ($S$) and fluorescent particle concentrations were investigated associated with INP concentrations (see Appendix A2). As discussed in

the previous section, we did not observe a significant difference in INP concentrations between autumn and spring. On the





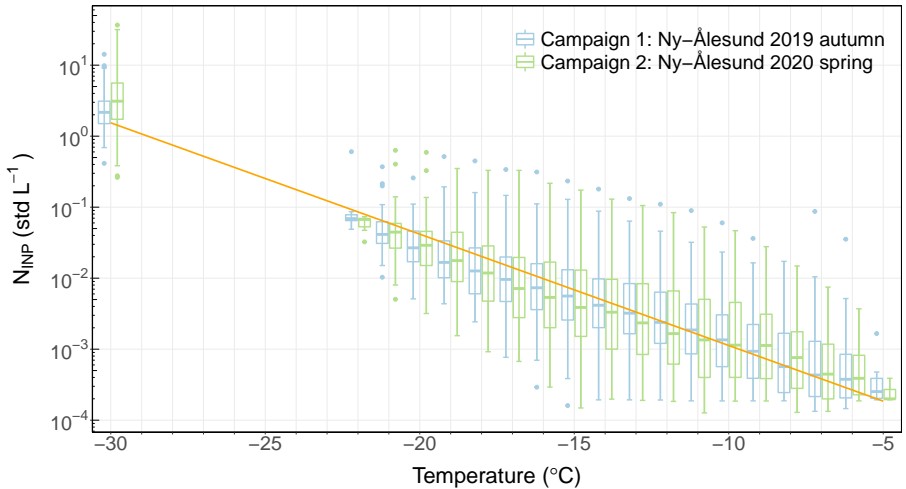

**Figure 3.** Box plot of observed INP concentrations as a function of temperature during the measurement campaigns in autumn 2019 (blue) and spring 2020 (green) in Ny-Ålesund, Svalbard. The orange line indicates a log-linear fit to all data combined.

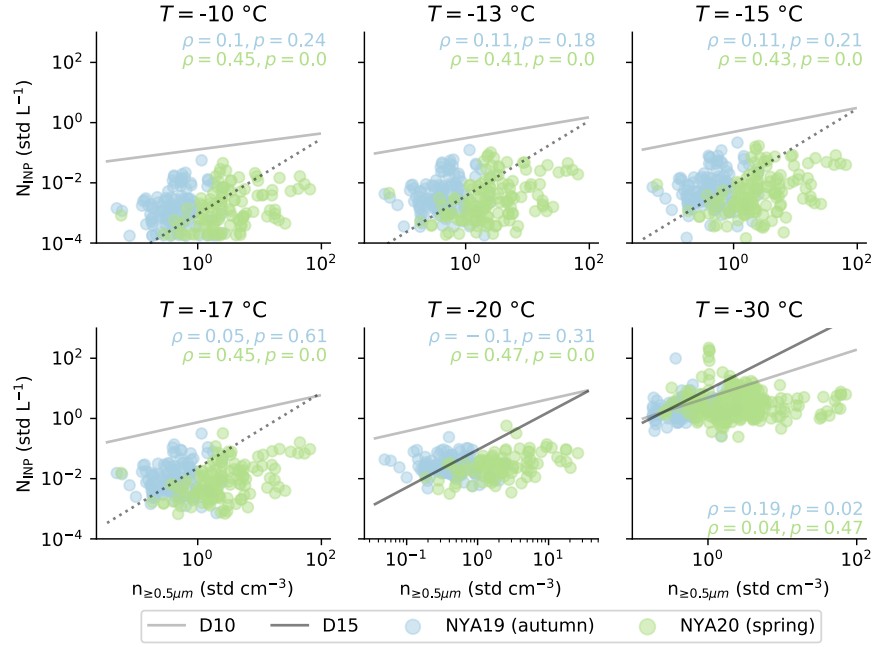

**Figure 4.** Observed INP concentration at selected temperatures as a function of the particle larger than $0.5\,\mu\text{m}$ (volume equivalent diameter) number concentration ($n_{\geq 0.5\mu\text{m}}$) during sampling in the autumn (blue) and spring (green) campaign. The Spearman's rank coefficient ($\rho$) and $p$ value are given for for each plot. Predicted INP concentration by DeMott et al. (2010) (D10) and DeMott et al. (2015) (D15) are presented in solid gray and black, respectively. Predictions of the parameterizations outside of the applicable temperature range are indicated in dashed lines.



other hand, the observed aerosol concentrations in spring were, on average, considerably higher than in autumn (see Appendix A1). This difference in aerosol burden between the seasons is not reflected in the observed INP concentrations, challenging the accurate prediction of INP concentration at low aerosol concentrations in the Arctic environment. In addition, in Fig 4, A3, and A4 the predictions for INP concentrations of a selection of existing parameterizations are shown. All parameterizations have

an aerosol component, giving them an output for any measured aerosol concentration. Three parameterizations (D10, N12, T13) overestimate, and one (M18) underestimates the INP concentrations. Note that the mentioned parameterizations represent air masses dominated by specific aerosol types different from our observations in remote Arctic regions. Since we observe insignificant aerosol concentration dependence for the INP concentration in both observed seasons, we propose to predict the INP concentration solely based on temperature.

**3.3 Log-normal distribution based INP parameterization**

In the derivation of parameterizations for INP concentrations, frequently, the relations were first fitted between the observed INP concentration and aerosol concentration (e.g., D10, D15) or surface area concentration (e.g., N12, M18) at a given temperature. Subsequently, the obtained relations were further linked with temperatures to develop a combined parameterization. This approach requires a robust relationship between INP concentration and aerosol number or surface concentrations which

was not evident in our in-situ observations. For a few decades, it has been known that the temperature dependence of INP parameterization is critical to represent cloud properties accurately (Hawker et al., 2021; Murray et al., 2021). We present a methodology to optimally fit the temperature dependence of INP concentration from frequency distributions.

Figure 5 shows the relative frequency distribution of our observed INP concentrations at different measured temperatures for the two campaigns in autumn and spring combined. The adequate log-normal fits of the observed distributions of INP

concentrations per temperature support the hypothesis of Ott (1990) (see Section 1). Figure A5 provides more evidence based on the approximate linearity between the observational and theoretical quantiles of the log-normal distribution, particularly at lower temperatures ($T < -12$ °C), where the closeness of data to the red line assesses the likelihood that the data set follows the theoretically log-normal distribution. Nevertheless, the trimmed tails of the distributions can be identified at higher temperatures ($T \geq -12$ °C), where INP concentrations are biased towards the minimum detectable concentration. The nature

of log-normal INP distributions has been previously reported from the long-term INP monitoring in Svalbard (Schrod et al., 2020) and the subtropical maritime boundary layer (Welti et al., 2018). Thus, we propose an INP parameterization that fits the median value of the log-normal distribution and temperatures. To account for challenges arising from instrumental limitations, we use the weighted least squares method (WLS, details in Section 2.2). To our knowledge, it is the first attempt to predict INP concentrations in the Arctic utilizing in-situ measurements. In Fig. 5, we compare to previous INP parameterizations that

are temperature dependent only (Fletcher et al., 1962; Cooper, 1986; Meyers et al., 1992). Our fit (fitting parameters given in Appendix A3) predicts at least one or two orders of magnitude lower INP concentrations. The INP concentration predicted by Schneider et al. (2021) is closer to our parameterization, yet still slightly overestimated.





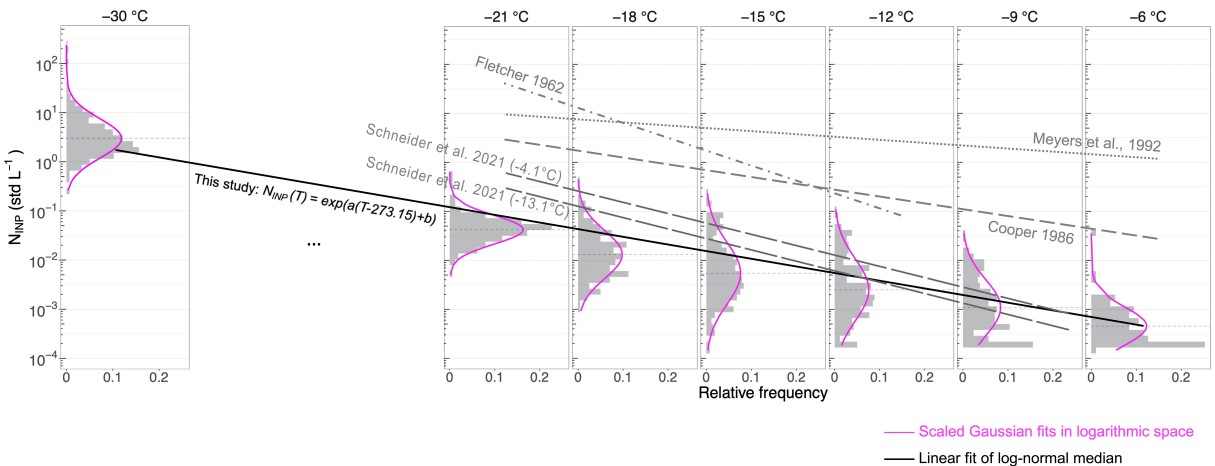

**Figure 5.** Relative frequency distribution of observed INP concentrations (gray histograms) for selected temperatures. Log-normal fit curves are presented for each histogram in magenta. The fit of all INP data in this study is presented in a solid black line (fitting parameters given in Appendix A3). Predictions from existing parameterizations inidcated by the gray lines. For Schneider et al. (2021), the temperature value in parentheses represent the mean ambient temperature during the observations in autumn 2019 and spring 2020, respectively, which were used to obtain the the INP predictions.

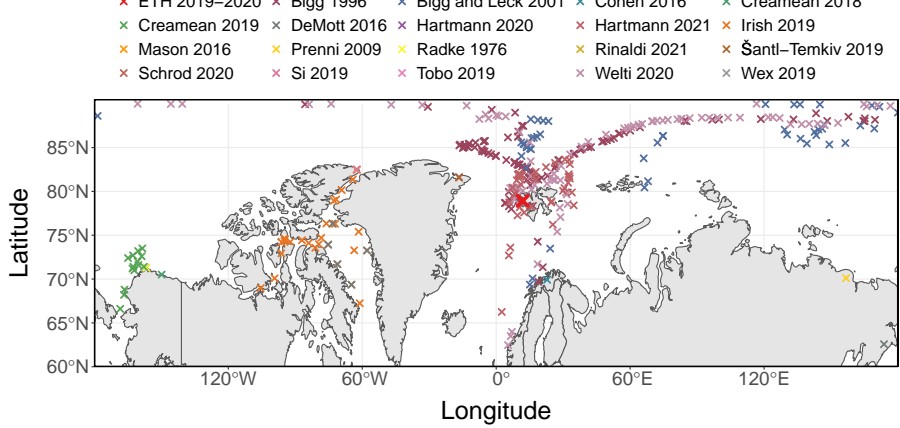

**Figure 6.** Locations of the previous Arctic field observations of INP concentrations used for evaluating the INP parameterization. More details of the observations of the individual campaigns are presented in Table A3.

### 3.4 Comparison to previous Arctic INP field observations

We gathered observations of INP concentrations from previous field measurements in the Arctic from 1976 to latest 2021 as a

reference data set to test the parameterization developed for background INP concentration in this study. Generally, INP data





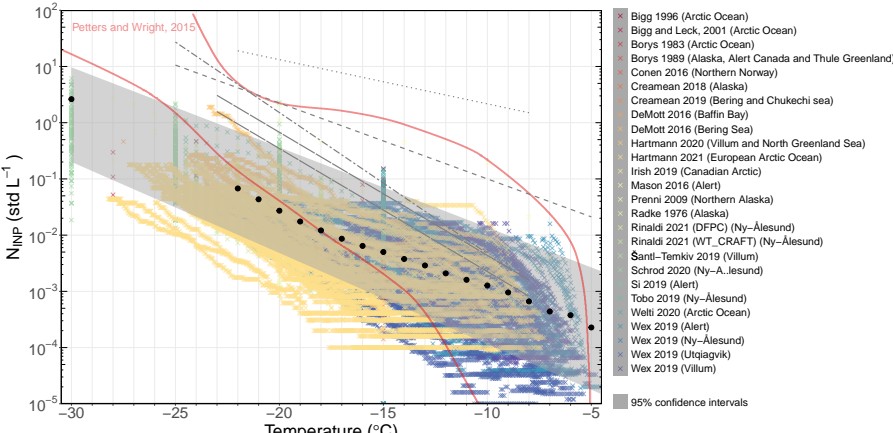

**Figure 7.** Comparison of INP concentrations measured in this study given as the log-normal median of the INP measurements in this study (black data point) with the grey shaded area representing the fit log-normal INP concentration distribution within the 95 % confidence interval. The area between two red lines are determined by Petters and Wright (2015) from precipitation samples. The dashed, dotted, dot-dashed, and long-dashed lines indicate the INP parameterization from Cooper (1986), Meyers et al. (1992), Fletcher et al. (1962), and Schneider et al. (2021) respectively. The temperature in parenthesis of the Schneider et al. (2021) parameterization represent the average ambient temperature observed during the autumn (-4.1 °C) and spring (-13.1 °C) campaigns.

that was clearly archived at locations further north than 66° 34′ were considered in this study (see positional information in Fig. 6, and detailed data features in Table A3). Overall, 32098 observations of INP concentration as a function of temperature were applied to evaluate the background INP parameterization developed from the Ny-Ålesund campaign data in this study. Figure 7 compares the measured INP concentrations in this study and the selected reference Arctic measurements. In general,

our probed INP concentrations range were in agreement with that reported in previous Arctic studies, with this study being one of the few to measure the INP concentrations in the Arctic at temperature as low as -30 °C. To evaluate our parameterization, we compared the predicted INP concentrations to observations from previous Arctic field measurements in Fig. 8. Regardless of the season, location, measuring instrumentation, and the nature of variation of INP concentrations, the trend of the INP concentrations is well captured by our parameterization. In Fig. 8a, from a total of 32098 comparison observations, 81 % of the

INP concentrations are predicted within the 95 % confidence interval, revealing notable predictability of the proposed Arctic parameterization, given that approximately 2 to 3 orders of magnitude of the variation of INP concentrations are naturally observed. However, the INP concentrations tend to be overestimated, particularly towards warm temperatures (T > -15 °C), because the INP concentrations are highly variable and rather low at these temperatures. More interestingly, the predictions of INP concentrations retain their performance when the evaluation is categorized for seasons (see Fig. 8b). Particularly in spring

and autumn, a higher percentage of observations from other studies fall into the 95 % confidence interval of the proposed parameterization. However, more INP concentrations are over-predicted in winter (Figure 8b, e.g., data from Wex et al. (2019)), particularly towards warmer temperatures, likely due to decreasing INP loading from regional emissions when the surface was





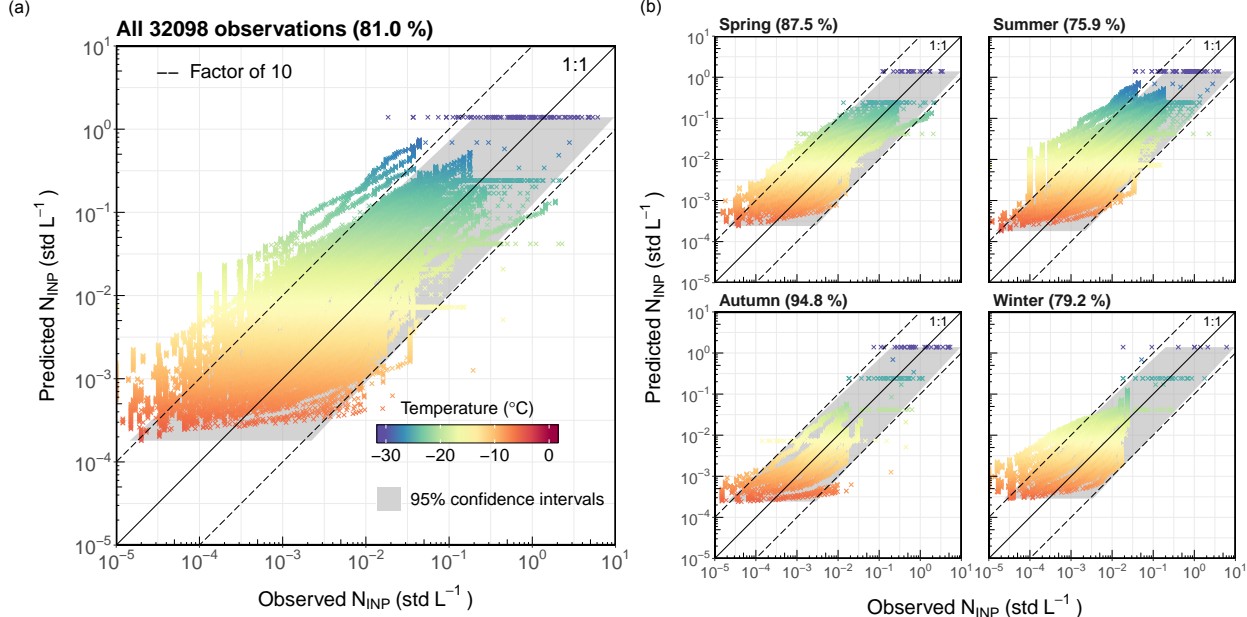

**Figure 8.** Predicted INP concentrations from the proposed fit (equation A1) compared to observations from previous Arctic field campaigns. An overview of the data used is given in Table A3. (a) All 32098 observational data. (b) Data per season (season classification: spring (March-May), summer (June-August), autumn (September-November) and winter (December-February)). The 1:1 line and a deviation of factor 10 are given in solid black and dashed black, respectively. Temperature of the corresponding INP concentration is given in color. The 95 % confidence interval of the fit is given in gray. The values in the parenthesis represent the percentages of predicted data falling within the 95 % confidence interval.

covered with more ice and snow. This finding agrees with our previous discussion that the LOD of DRINCZ imposed limitations fitting the low INP concentrations at warm temperatures, notably when overall INP loading declines and the log-normal distributions become biased to the higher INP concentrations that were observed above the LOD. Based on this explanation, the performance of the parameterization should improve in the summer when overall INP concentrations increase due to the increasing local emissions (Tobo et al., 2019; Creamean et al., 2019), which is, however, contradictory to what is shown in Fig. 8b. The predicted INP concentrations are still overestimated, mainly due to the inclusion of 11804 data points from Hartmann et al. (2021), who observed universally lower INP concentrations during a ship-based campaign in the Arctic ocean around Svalbard compared to the provided distribution (i.e., 95 % confidence interval) predictions. They measured potentially in a region where comparably lower INP concentrations were prevailing, i.e., when measured within sea ice pack, INP concentrations were systematically lower compared to the ice-free ocean. Another possibility could be the degradation of the filter samples during transport and storage, supported by the fact that the INPs measured by SPIN (the online instrument) were consistently higher than filter samples measured by their drop freezing method, LINA (Hartmann et al., 2021). The above reasons could explain why the data from Hartmann et al. (2021) are substantially lower than previous measurements. If Hartmann et al. (2021)





data is removed from the evaluation, we achieve approximately 97 % of the data falling within the confidence interval for the summer (not shown).

## 4    Conclusions and atmospheric implications

A 12-week field measurement campaign on ambient INP number concentrations and aerosol properties was undertaken in au-
tumn 2019 and spring 2020 in the Norwegian Arctic in this study. Based on a random dilution model (Ott, 1990), the measured INP concentrations naturally converge to a log-normal frequency distribution if the INP originated from a mix of locally and long-ranged sources. During the measurement periods, no significant relationship was observed between the INP concentrations and physical aerosol parameters. Therefore, we developed a log-normal-distribution-based parameterization to predict the median and variation ($2\sigma$) of atmospheric background INP concentrations dependent solely on temperature. The new parame-
terization was compared to INP concentrations observed by previous Arctic field measurements, and in general, demonstrated promising predictability within the 95 % confidence interval, although deviations are larger towards warm temperatures. Note that the presented INP parameterization is specified for the Arctic environment, where the atmosphere is well-mixed and transient effects average out, and is easy to implement given the simplified form. We hope future modeling studies will test the sensitivity of the given parameterization and its effects on cloud properties. The presence of well-mixed INP air masses is
exhibited by the absence of a relationship with aerosol properties and further by the inability of previous aerosol-based INP parameterizations to reproduce the observations from this study. The new INP parameterization can be used as a proxy to estimate the pre-industrial or pristine INP level, and it can be applied to research related to cloud properties as modeling results showed that the Arctic MPCs respond actively to the INP perturbations (Eirund et al., 2019), and Arctic amplification was enhanced given large and fewer ice particles in Arctic MPCs (Tan and Storelvmo, 2019). Our INP parameterization promotes
future modeling studies via a more realistic microphysical representation in the Arctic MPCs, especially the vertical profile of primary ice distribution (Hawker et al., 2021), thus, improving the predictions for the future Arctic climate.

## Appendix A

### A1    Time series for particle number concentration larger than 0.5 µm

Figure A2 shows the aerosol number concentration for particles with physical diameter larger than 0.5 µm as function of the
sample number. Each sample lasted for three minutes for a total of 20828 and 18966 samples, from October 5–November 18, 2019 for the autumn campaign and from March 15–April 24, 2020 for the spring campaign, respectively. Observed median concentrations were 0.43 std $\mathrm{cm}^{-3}$ and 2.51 std $\mathrm{cm}^{-3}$ during the autumn and spring campaign, respectively.

### A2    INP concentration related to other aerosol properties

Biological particles are found to be one of the more prominent contributors to INP concentrations at warmer temperatures
(T > −15°C) (Kanji et al., 2017 and references therein). During the campaign in autumn, the concentration of fluorescent





particles indicative of biological particles was measured in addition to the size and surface of the ambient aerosol. Generally, a weak correlation between INP concentration and concentration of fluorescent particle concentration (see the Spearman's rank coefficients per temperature in Figure A4). Given the increased correlation to the size properties in spring (see Figures 4 and A3), it is conceivable to expect the relation between INP concentration and concentration of fluorescent particle concentration

also to be stronger during spring.

### A3 Log-normal distribution based INP parameterization

The INP parameterization based on log-normal distribution represented as:

$$N_{\mathrm{INP}}(T) = e^{a(T-273.15)+b}, \tag{A1}$$

  Where $N_{\mathrm{INP}}$ is the INP number concentrations in std $\mathrm{L}^{-1}$, $T$ is the temperature in Kelvin, and $a$ and $b$ are fit parameters

with the value given in Table A2. Note that instead of using an OLS based linear regression, we applied a WLS approach (see Appendix 2.2) to address the heteroscedasticity of frequency distributions of INP concentrations over different temperatures. In addition, we also report the dominant INP concentration distributions within the 95 % confidence interval (i.e., median $\pm$ 2$\times$standard deviation), based on the WLS fit of log-normal distributions. The fitting parameters for the lower and upper bounds of predicted INP concentration distribution are also given in Table A2.

**Table A1.** List of parameters used for linear model fit evaluation. WLS_W$_\sigma$, WLS_W$_{\mathrm{obs}}$ and WLS_W$_{\sigma^2}$ represent different weighing factors of the WLS linear regression model, i.e., standard deviation, number of observations at each measured temperature, and variance at each measured temperature, respectively. RMSE and MAPE symbolize the lowest root-mean-square error and mean absolute percentage error, respectively. The highlighted WLS method was selected to develop the INP parameterization in this study.

| Fitting method | $r^2$ | RMSE | MAPE (%) |
|---|---|---|---|
| OLS | 0.9774 | 6.5708 | 2.2249 |
| WLS_W$_\sigma$ | 0.9743 | 6.5791 | 2.3112 |
| **WLS_W$_{\mathbf{obs}}$** | **0.9778** | **6.5408** | **2.0302** |
| WLS_W$_{\sigma^2}$ | 0.9721 | 6.5899 | 2.4352 |





**Table A2.** List of parameters for for the proposed INP concentration parameterization (Equation A1). Median, and lower and upper 95 % CI represent the parameters in Equation A1 for the median, and lower and upper bound of 95 % confidence interval, respectively, of the linear fit for the log-normal distribution.

| Fitting parameter | a | b |
|---|---|---|
| Median | -0.3504 | -10.1826 |
| Lower 95 % CI | -0.3731 | -12.7993 |
| Upper 95 % CI | -0.3278 | -7.5659 |





**Table A3.** List of INP measurement campaigns with associated data in the Arctic. The "Platform" column includes the ground-based (GB), ship-borne (SB) and airborne (AB) measurement. The reported sampling technique involves filter, impactor and continuous flow diffusion chamber (CFDC); and the INP analysis method are droplet-freezing (DF), thermal diffusion chamber (TDC) and CFDC.

| Reference | Observation NO. | Location | Platform | Latitude (°) | Altitude (m a.s.l) | Sampling technique | INP analysis | Sampling time | Volume (m³ air) |
|---|---|---|---|---|---|---|---|---|---|
| [1] | 2271 | Ny-Ålesund, Svalbard | GB | 78.9 N | 11 | Filter | DF | Mar 2012 - Sep 2012 | 213.3 |
| [1] | 5357 | Alert, Canada | GB | 82.5 N | 210 | Filter | DF | Apr 2015 - Apr 2016 | 16792.8 |
| [1] | 4734 | Utqiagvik, Alaska | GB | 71.3 N | 11 | Filter | DF | Jan - Aug 2013 | 23984.3 |
| [1] | 1323 | Villum, Greenland | GB | 81.6 N | 24 | Filter | DF | Jan - Dec 2015 | 5000.0 |
| [2] | 298 | Ny-Ålesund, Svalbard | GB | 78.9 N | 475 | Filter | DF | Jul 2016 and Mar 2017 | 15.6 |
| [3] | 96 | Ny-Ålesund, Svalbard | GB | 78.9 N | 71 | Filter | TDC | Apr 2018 and Jul 2018 | 9.2 |
| [3] | 622 | Ny-Ålesund, Svalbard | GB | 78.9 N | 71 | Filter | DF | Apr 2018 - Aug 2018 | 864.0 |
| [4] | 4 | Prudhoe Bay oilfield, Alaska | GB | 70.5 N | 2 | Impactor | DF | Mar 2017 - May 2017 | 38.4 |
| [5] | 3 | Canadian Arctic | SB | 67.2 - 81.4 N | 15 | Impactor | DF | Jul 2014 - Aug 2014 | 0.2 |
| [6] | 22 | Alert, Canada | GB | 82.5 N | 200 | Impactor | DF | Mar 2014 - Jul 2014 | 32.4 |
| [7] | 5 | Bering sea | SB | 62.6 N | 20 | Filter | DF | Jul 2012 | 13.6 |
| [7] | 3 | Baffin Bay | SB | 62.6 - 76.3 N | 20 | Filter | DF | Jul 2014 | 0.2 |
| [8] | 152 | Arctic Ocean | SB | 75.0 - 90.0 N | 25 | Filter | TDC | Jul 1991 - Oct 1991 | 3.0 |
| [9, 20] | 305 | Arctic Ocean | SB | 69.4 - 89.9 N | 25 | Filter | TDC | Jul 1996 - Sep 1996 | 0.6 |
| [10] | 34 | Alaska | GB | 71.3 N | 0 | Filter; CFDC | DF; CFDC | Mar 1970 | 3.0 |
| [11] | 27 | Northern Norway | GB | 69.9 N | 907 | Filter | DF | Jul 2015 | 24.0 |
| [12] | 2 | Alaska, Alert and Greenland | AB | - | - | Filter | TDC | Apr 1986 | 1.4 |
| [13] | 15 | Arctic Ocean | SB | - | - | Filter | TDC | Summer and winter 1980 | 1.4 |
| [14] | 38 | Alert, Canada | GB | 82.5 N | 185 | Impactor | DF | Mar 2016 | 1.2 |
| [15] | 2194 | Villum and North Greenland Sea | AB | 81.6 N | - | Filter | DF | Mar 2018 - Apr 2018 | 4.3 |
| [16] | 18 | Villum, Greenland | GB | 81.6 N | 24 | Filter; impinger | DF | Apr and Aug 2016 | 21-75; 145-800 |
| [17] | 14 | Bering and Chukechi sea | SB | 66.6 - 73.5 N | 20 | Impactor | DF | Aug - Sep 2017 | 39.71 |
| [18] | 1038 | Ny-Ålesund, Svalbard | GB | 78.9 N | 474 | Electrostatic precipitator | CFDC | May 2015 - Jan 2017 | - |
| [19, 20] | 12688 | European Arctic Ocean | SB | 66.3 - 83.7 N | - | Filter | DF | May - Jul 2017 | 3.3 - 13.9 |
| [20] | 825 | Arctic Ocean | SB | 62.5 - 90.0 N | - | Filter | DF | Jul - Aug 2001 | - |
| [21] | 10 | Northern Alaska | AB | - | - | CFDC | CFDC | Jul - Oct 2004 | 0.06 (per hour) |

[1]: Wex et al. (2019); [2]: Tobo et al. (2019); [3]: Rinaldi et al. (2021); [4]: Creamean et al. (2018); [5]: Irish et al. (2019); [6]: Mason et al. (2016); [7]: DeMott et al. (2016); [8]: Bigg (1996); [9]: Bigg and Leck (2001); [10]: Radke et al. (1976); [11]: Conen et al. (2016); [12]: Borys (1989); [13]: Borys and Grant (1983); [14]: Si et al. (2019); [15]: Hartmann et al. (2020); [16]: Šantl Temkiv et al. (2019); [17]: Creamean et al. (2019); [18]: Schrod et al. (2020); [19]: Hartmann et al. (2021); [20]: Welti et al. (2020); [21]: Prenni et al. (2009).

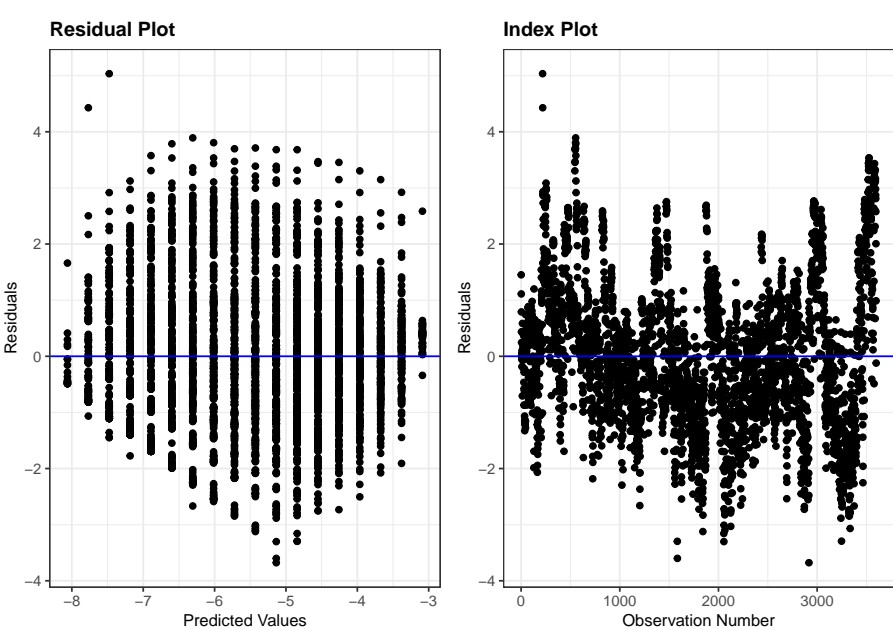

**Figure A1.** Left: residual distribution along the predicted values (logarithmically transformed INP concentrations). The predicted values are the power of base $e$. Right: residual distribution according to the index of observations.

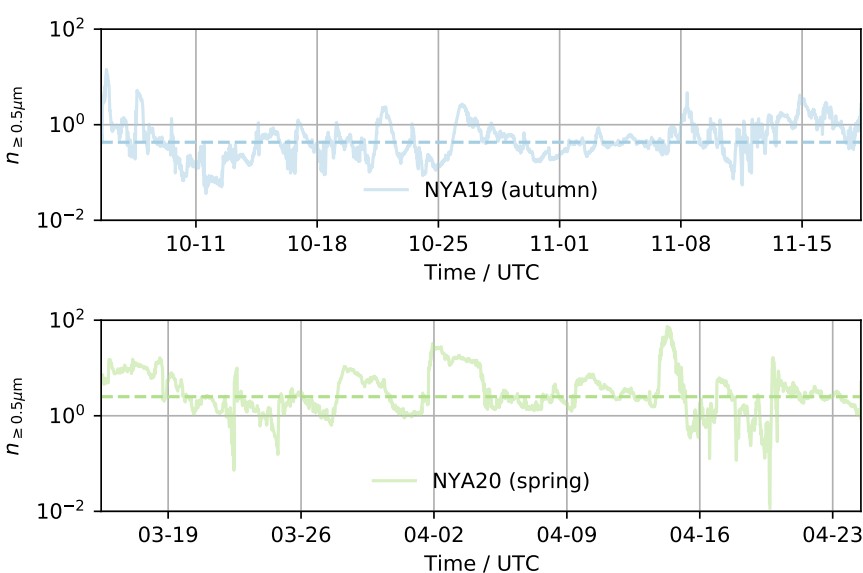

**Figure A2.** Time series of particle larger than 0.5 µm physical diameter number concentration for the autumn (blue, upper panel) and spring (green, lower panel) campaign. Median concentrations for each season are indicated by the dashed horizontal lines.



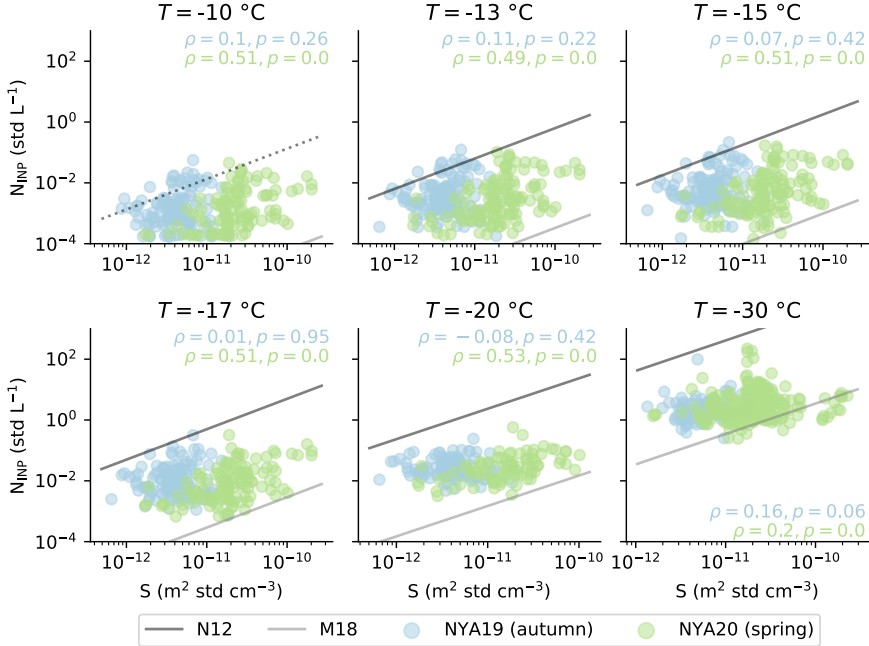

**Figure A3.** Observed INP concentration at selected temperatures as a function of the particle surface concentration $S$ during sampling in the autumn (blue) and spring (green) campaign. The Spearman's rank coefficient ($\rho$) and $p$ value are given for each plot. Predicted INP concentration by Niemand et al. (2012) (N12) and McCluskey et al. (2018) (M18) are presented solid black and gray, respectively. Predictions of the parameterizations outside of the applicable temperature range are indicated in dashed lines.



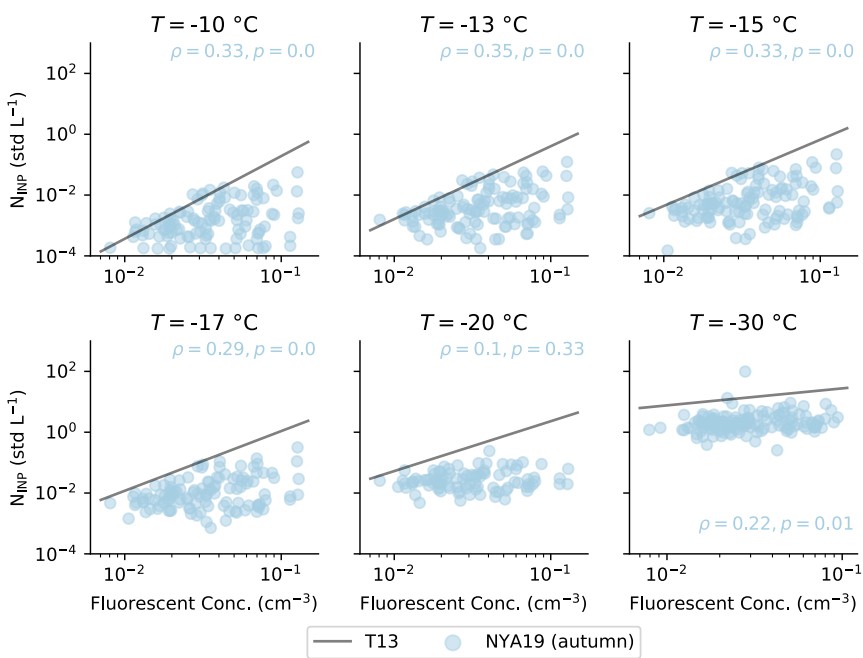

**Figure A4.** Observed INP concentration at selected temperatures as a function of the fluorescent particle concentration present during sampling for the autumn (blue) campaign. The Spearman's rank coefficient ($\rho$) and $p$ value are given for each plot. Predicted INP concentration by Tobo et al. (2013) (T13) is presented with the solid black line.



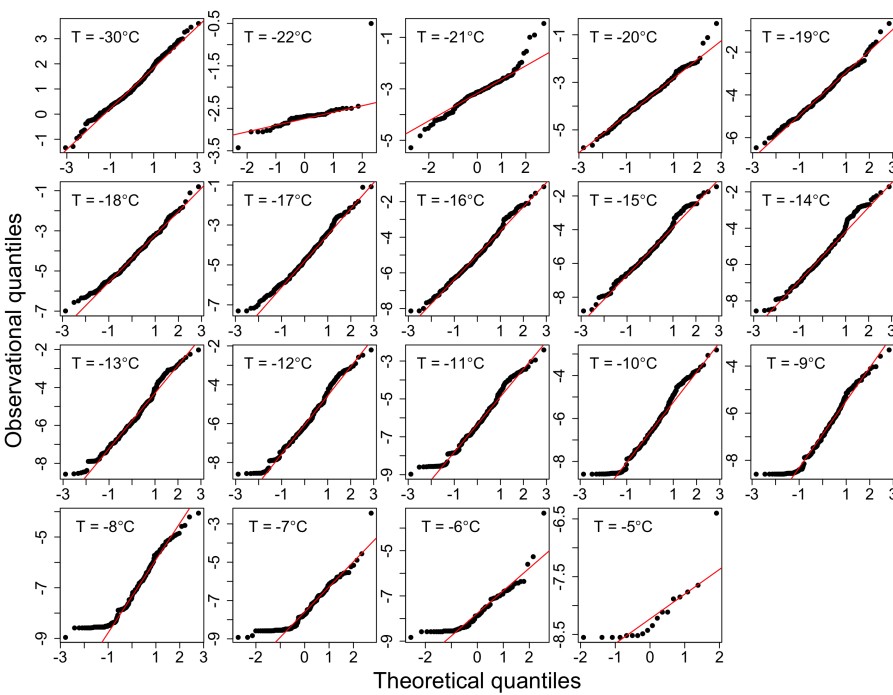

**Figure A5.** QQ-plot (quantile-quantile plot) for INP concentrations measured at different temperatures. The value on both axes represent the exponential power to the base of $e$. The red solid lines are theoretical references according to the log-normal distribution.



*Data availability.* The data generated and analyzed in this study will be made available from a publicly accessible ETH repository. Note by authors: data will be uploaded upon acceptance of publication.

*Author contributions.* GL and JW contribute equally to this study. GL and JW performed the INP and aerosol measurements, analyzed the data and prepared the figures. ZAK conceived the idea of the parameterization. GL, JW and ZAK interpreted the data. GL and JW drafted the manuscript with contributions from ZAK. JTP and JH were involved in conceiving and organizing the field study. All authors reviewed

and commented on the manuscript.

*Competing interests.* The authors declare that no competing interests are present.

*Acknowledgements.* GL and ZAK acknowledge that this project has been made possible by a grant of the Swiss Polar Institute, Dr. Frederik Paulsen. JW, JTP and JH acknowledge the Swiss National Science Foundation (SNSF) (grant NO. 200021_175824), European Union's Horizon 2020 research and innovation program (grant NO. 821205), and the Swiss Polar Institute (Exploratory Grants 2018) for funding. We

are grateful for their financial support. We acknowledge all those involved in the field work associated with the NASCENT project, including technical support from Dr. Michael Rösch, Dr. Robert O. David, and from the AWIPEV and Norwegian Polar Institute. We would like to thank Dr. Keith Bigg and Dr. André Welti for sharing their research data. We want to express our deepest gratitude to Maxim Samarin for invaluable discussions regarding the temperature fit.





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
