# Peer review of "Predicting atmospheric background number concentration of ice nucleating particles in the Arctic"

_Atmospheric Chemistry and Physics, 2022_

## Author Comment (AC1)

**Referee comments 1**

We thank referee 1 for the valuable feedback on our manuscript acp-2022-21. In response to the questions and suggestions, please find our answers and revisions listed below. **Referee comments are reproduced in bold** and author responses in normal font; *extracts from the original manuscript are presented in red italic* and *extracts from the revised manuscript in blue italic*.

**Review of "Predicting atmospheric background number concentration of ice nucleating particles in the Arctic" by Li et al., submitted to ACPD**

**In this study, data from two measurement periods in Ny-Alesund (Svalbard) are used. Collected data concern mainly INP concentrations (both in-situ and off-line analysis), as well as some aerosol properties (size distributions for both seasons, fluorescence signals for one season). It is argued that INP concentrations cannot be described well based on the aerosol properties, and a new INP parameterization is derived from the data, given as a line, similar to the Fletcher or the Cooper curve, together with a 95% confidence interval (covering roughly one order of magnitude).**

**The work as such is sound. However, I have a number of issues with both content and formulations, which I elaborate on below. Besides for three main concerns, there is a number of general comments, followed by editorial comments. The three main concerns all center around how broadly applicable the new parameterization really is, due to the time periods during which measurements were done, due to the fact that parameterizations based on aerosol properties were not done (it was merely compared with older parameterizations for other locations, which is not enough to reject these kinds of parameterizations) and about some data which may have been overlooked in an analysis connecting the new parameterization to literature data (SPIN data from Hartmann et al., 2021, a data source of which, however, off-line data was used). Also, overall, tuning down some of the statements made in the text would make the work more scientifically sound.**

**Overall, this is an interesting study which will add to the community's understanding of Arctic INP, and it certainly can be revised such that it will merit publication. However, (rather major) revisions are needed.**

We thank the reviewer for taking the time to give us detailed comments. We revised the manuscript accordingly and respond to each of the specific comments below.

**Three major issues:**
**One of my main concerns is about the time periods during which you measured. You happen to have measured during times which were described as transition periods from low to high (spring) or high to low (fall) INP concentrations before (e.g., Creamean et al. 2018; Wex et al. 2019). I am aware that Schrod et al. (2020a) did not find such an annual variation, but they only examined temperatures < -20°C. The pronounced changes shown in Creamean et al. (2018) and Wex et al. (2019) occurred at higher temperatures, for which you report the bulk of your data. Also, there is a new publication which you may not know yet (Porter et al., 2021), in which INP concentrations in the North Pole region were measured in summer. The there reported INP concentrations are MUCH higher than yours (by roughly close to 3 orders of magnitude), which means that while you overestimate concentrations in winter, compared to these new data, you will underestimate them in summer. (And while Porter et al. (2021) is similar to a discussion paper in ACPD, I am sure it will be published in peer reviewed version soon, and then your statements about a general applicability of your fit will be outdated already from its start.) Also, Tobo et al. (2019) and Sanchez-Marroquin et al. (2020) suggested that Arctic mineral dust sources contribute to atmospheric INP.**

We thank the referee for the feedback. We agree with the referee that higher INP concentrations were previously observed, especially in summer when the surface is free of ice and snow cover; thus, pronounced local sources (e.g., dust, biological INPs) are expected. Despite the importance of augmented local sources in summer, only a few Arctic INP measurement campaigns focused on transition seasons (spring and autumn),

which is what this study focuses on. In this context, no seasonal variation is observed in our measurements. Indeed, we acknowledge the limitation that our parameterization is developed from the transition seasons and could pose a bias when simply applied to periodical studies, e.g., predicting INP concentrations in summer or winter. However, the distribution-based parameterization provides useful information on the general level and range of INP concentrations in the Arctic environment, which could be applied to regional climate models for long-term predictions. Indeed, measurements during transition seasons in the Arctic are rare in the literature, where our study fills in the gap. It is noteworthy that when considering the transition seasons, there is no seasonal influence and this, we believe, would be a valuable contribution to the existing literature which presents seasonal influences between summer and winter.

In addition, we think that the extent of seasonal/temporal variation that could be observed strongly depends on the sample size and resolution. Take the INP measurements from Porter et al., 2021 as an example, a total of around 30 filter samples were collected for INP analysis on an approximately daily basis, and only three of them had substantially high INP concentrations compared to the average level we observed. An instant INP enhancement from local sources, e.g., wind erosion carrying mineral dust particles, could easily enhance the average INP concentration of the filters of 1-day resolution. One advantage of our measurements was that by combining the high-resolution HINC data (quasi-continuous, 20 minutes) and DRINCZ measurement from the high flow rate impinger samples (quasi-continuous, 1 hour), we were able to obtain a large data set containing over 4000 observations over 12 weeks. Furthermore, developing a distribution-based parameterization benefits from these high-frequency data by preventing local enhancement from instantaneous cases. Individual measurements were not shown in the article, but it is visible in Figure 3 of our initial manuscript that we also observed several peak cases when there were striking INP concentrations comparable to the 3 peak samples observed by Porter et al., 2021. In other words, our log-normal distribution based parameterization is insensitive to peak events and therefore is recommended to be implemented for longer span predictions.

In this regard, several sentences are added to discuss the discrepancy in line 216 (updated version from line 261): "*However, a few studies (e.g., Wex et al., 2019, Tobo et al., 2019, Creamean et al., 2018) observed consistently higher INP concentrations in the summer, indicating enhanced sources of local emissions due to the decreased ice cover. We note that the parameterization herein was derived from the measurements during transition seasons (autumn and spring), aiming to predict the background level of INP concentrations in the Arctic. Therefore, applying it to generate INP concentrations, particularly in the Arctic summer, could introduce a low bias. On the other hand, our log-normal distribution based parameterization developed from high-frequency INP measurements (over 4000 observations in 12 weeks) is insensitive to peak events, i.e., local INP enhancement from instantaneous cases. It is therefore recommended to be implemented for longer-span predictions.*"

We also modified the abstract to address the absence of a seasonal difference and the transition seasons (see lines 10-15 revised manuscript), as well as address this in the conclusions (see line 314ff revised manuscript).

**It is unknown but possible that average land based INP concentrations may differ from averages of data taken over the ocean. This should be discussed more.**

We agree with the reviewer's comment and therefore had a look into the literature if previous campaigns measured differences between measurements taken over land and ocean. Overall, we did not find a strong dependence of measured INP concentrations on the location of observations, i.e., ground-based, airborne, and ship-based. Further details are addressed under the comment on Line 269ff.

**2) The second of my main concerns is your suggestion that a single line (together with a confidence interval) will represent INP concentrations better than a parameterization based on aerosol properties. With this, I do not mean to say that such a parameterization based on aerosol properties will work out. But judging**

**from what I see from your data in Fig. 4, A3 and A4, doing fits for n>0.5mm, S or fluorescent conc. for your dataset would cause a similar uncertainty (or confidence interval) of one order of magnitude as your temperature dependent parameterization. And as such, aerosol parameters may represent the data just as well as the parameterization you present. (While, admittedly, additional measurements are needed to retrieve them, so I get the nice part of your approach.)**

We thank the referee for this comment and for recognizing the advantage of the simplicity of the proposed parameterization. For aerosol property parameters $n_{0.5}$, S and fluorescent concentrations as predictors for INP concentrations (Figure 4, 5, S4 and S5 in the revised manuscript), the mentioned parameterizations (D10, D15, N12, M18 and T13) represent air masses dominated by specific aerosol types, which are likely unrealistic for our observations in the Arctic region. In terms of predictability of INP concentrations, indeed, a similar uncertainty range was observed using the suggested aerosol property parameters compared to our temperature-dependent parameterization, which could be attributed to the inherent variation of naturally occurring INP concentrations covering a broad range. The aerosol-property parameterization fits deviate from our measured INP concentrations almost at all temperatures (except that D15 produced a good prediction at -20 °C). Therefore, our temperature-dependent parameterization outcompetes the aerosol-property-based parameterizations in predicting INP concentrations at all observed temperatures. Additionally, no strong correlation between INP concentrations at different nucleation temperatures and aerosol property parameters was observed for data of both seasons combined (Table S1 in the supplementary information), indicating the need for developing a parameterization independent of these parameters.

**Instead for examining also these other types of parameterizations in detail, you are just showing parameterizations from totally different environments and then turn them down on false claims. Obviously also the line fits from other environments (Fletcher, Cooper, Meyers) similarly do not fit. So you need to change your arguments for not using an aerosol parameter based parameterization.**

**Therefore, it needs to be discussed why you think that such a temperature dependent parameterization would outperform parameterizations related to aerosol parameters, which, as said, aren't even introduced and much less examined in the current study. Also, while your parameterization gives a general temperature dependent trend, it does not reproduce the variability as such (besides for giving a range for it, which covers, overall, 2 orders of magnitude). All of this needs to be stated much clearer.**

We thank the referee for the comment and for pointing out the need for a more detailed argumentation. During our analysis, we constructed fits for every aerosol property investigated and evaluated their performance. However, the improvement in predictability compared to the temperature fit was low (Spearman's correlation < 0.4) or insignificant as shown in Table S1 (revised manuscript and see below). If significant slopes of most fits (exponent of the power-law functions) were considerably low, adding nearly no additional dependence to the parameter in question. Furthermore, and more importantly, as described earlier in the manuscript, no seasonal difference in INP concentration was observed (Fig. 3) while a seasonal difference was observed for the aerosol properties (Figure S2 and see two clusters per season in Figs. 4 and 5). Therefore, the use of an aerosol property for an INP prediction (which would only be advisable if the slope is greater than zero) would in our case predict higher INP concentrations due to the higher aerosol loading in spring and vice versa in autumn, which would induce a seasonal bias that was not observed. In the revised manuscript, we expanded our argumentation, introduced a new Figure 5, added Table S1 to the Supplementary Information, and adapted Section 3.2 to:

[Figure]

**Figure 5.** Observed INP concentration at a nucleation temperature of $-15$ °C ($N_{INP}^{-15°C}$) as a function of (a) the ambient aerosol surface area concentration ($S$), and (b) the ambient number concentration of fluorescent particles during sampling in the autumn (violet) and spring (green) campaign. The green points are absent in (b) because the fluorescence measurements were only conducted in autumn campaign. The Spearman's rank coefficient ($\rho$) and $p$-value are given for each season in each plot (the correlation for both seasons combined is presented in Table S1 in the Supplementary Information). Further indicated are predicted INP concentrations by Niemand et al. (2012) (N12), McCluskey et al. (2018) (M18), and Tobo et al. (2013) (T13).

**Table S1.** Spearman's correlation coefficients between INP concentration at different nucleation temperatures to aerosol and ambient parameters: Total aerosol concentration ($n_{tot}$), aerosol concentration of particles with diameter larger 0.5, 1.0, and 2.0 µm ($n_{0.5}, n_{1.0}, n_{2.0}$), aerosol surface area concentration (S), concentration of fluorescent particles (Fluorescent conc.), ambient ground temperature ($T_{amb.}$), virtual ground temperature ($T_v$), potential ground temperature ($\theta$), equivalent potential ground temperature ($\theta_E$), ambient ground relative humidity ($RH_{amb.}$), ambient ground pressure ($p_{amb.}$), ground wind direction (wd), ground wind speed (ws). Coefficients in bold represent a significant relation (significance level $p < 0.05$).

| Factor | $-6$ °C | $-8$ °C | $-10$ °C | $-12$ °C | $-14$ °C | $-16$ °C | $-18$ °C | $-20$ °C | $-30$ °C |
|---|---|---|---|---|---|---|---|---|---|
| $n_{tot}$ (std cm$^{-3}$) | 0.043 | 0.113 | **0.121** | 0.087 | 0.101 | 0.117 | **0.161** | 0.124 | **0.215** |
| $n_{0.5}$ (std cm$^{-3}$) | 0.049 | 0.117 | **0.126** | 0.093 | 0.109 | **0.126** | **0.167** | 0.127 | **0.202** |
| $n_{1.0}$ (std cm$^{-3}$) | 0.082 | **0.154** | **0.168** | 0.138 | **0.155** | **0.167** | **0.194** | 0.121 | **0.213** |
| $n_{2.0}$ (std cm$^{-3}$) | **0.171** | **0.265** | **0.282** | **0.259** | **0.272** | **0.276** | **0.277** | **0.173** | **0.266** |
| $S$ (m$^2$ std cm$^{-3}$) | 0.04 | **0.128** | **0.148** | **0.122** | **0.13** | **0.148** | **0.198** | **0.175** | **0.307** |
| Fluorescent conc. (cm$^{-3}$)* | **0.355** | **0.311** | **0.328** | **0.321** | **0.338** | **0.273** | **0.28** | 0.102 | **0.22** |
| $T_{amb.}$ (°C) | $-0.021$ | $-0.059$ | $-0.065$ | $-0.031$ | $-0.05$ | $-0.093$ | **$-0.168$** | **$-0.289$** | 0.063 |
| $T_v$ (K) | $-0.025$ | $-0.067$ | $-0.071$ | $-0.047$ | $-0.057$ | $-0.096$ | **$-0.179$** | **$-0.282$** | 0.063 |
| $\theta$ (K) | $-0.031$ | $-0.057$ | $-0.066$ | $-0.034$ | $-0.049$ | $-0.091$ | **$-0.164$** | **$-0.292$** | 0.055 |
| $\theta_E$ (K) | $-0.031$ | $-0.065$ | $-0.071$ | $-0.042$ | $-0.054$ | $-0.095$ | **$-0.172$** | **$-0.294$** | 0.053 |
| $RH_{amb.}$ (%) | 0.005 | $-0.031$ | $-0.03$ | $-0.057$ | $-0.017$ | $-0.024$ | $-0.091$ | $-0.096$ | **$-0.137$** |
| $p_{amb.}$ (hPa) | **0.121** | 0.061 | 0.091 | 0.105 | 0.061 | 0.062 | 0.007 | 0.042 | **0.172** |
| wd (°) | 0.044 | 0.014 | 0.016 | $-0.028$ | $-0.031$ | 0.012 | $-0.043$ | $-0.005$ | **0.094** |
| ws (m/s) | **0.245** | **0.367** | **0.37** | **0.366** | **0.413** | **0.433** | **0.353** | **0.204** | **0.118** |

* Note that the fluorescent particle concentration was only available for the autumn data and hence the presented correlation coefficients are restricted to autumn.

[revised manuscript text omitted]

**3)**
**My last main concern is about your comparison with literature data. In Fig. 7, you include some data from Hartmann et al. (2021), a ship-based summer campaign. More specifically, you include the off-line data in Fig. 7, but not the SPIN data. Later in the text you argue that the off-line data are lower than yours, and**

**you mention as possible reasons a) local differences and b) that the off-line samples may have degraded during transport and storage, based on an extrapolation of these data towards SPIN data shown in Hartmann et al. (2021). Specifically, your argument b) makes me think that you feel that the SPIN data is more trustworthy than the off-line data. But this opens up the question why SPIN data is not included in Fig. 7.**

**Then you say that without including the Hartmann et al. (2021) data, "approximately 97 % of the data falling within the confidence interval for the summer." I assume that for your analysis comparing with literature data, you only included the off-line data from Hartmann et al. (2021) shown in Fig. 7. Or were the SPIN data ever included in your comparison to previous Arctic INP field observations? I guess not, and this is disturbing. I assume you overlooked to include these data and will do so in the revised version. Or maybe there is a good reason for excluding these data. In this case, it needs to be explicitly stated in the text that you exclude them and why.**

**Summarizing these major concerns, claims you make about the wide applicability of your new parameterization and its outperformance of others seem to be exaggerated and need to be tuned down. I refer to this issue below explicitly where needed.**

We thank the referee for pointing out the problem. The referee stated correctly that the SPIN data was not used in evaluating our parameterization in the previous version, but we compared them to the offline data in the discussion as we erroneously assumed that the SPIN data were in there. In the updated version, the SPIN data are included in Figure 8 with clearer labels and legends (shown below with the legend "Hartmann 2021 (European arctic Ocean)"), and the INP concentrations measured by SPIN were consistently higher compared to our predicted ranges and offline data from Hartmann et al., 2021. However, in Figure 10 (Figure 8 in original version), we exclude SPIN data measured at nucleation temperatures lower than -30 °C to evaluate our parameterization because -30 °C was the lowest temperature at which we measured.

[Figure]

**Figure 8.** Comparison of Arctic INP concentration measurements in different studies. The INP parameterization developed from this study is presented in a solid black line, with the grey shaded area representing the log-normal INP concentration distribution within the 95 % confidence intervals. The area between two red lines is a compilation of INP concentrations determined from precipitation samples from the mid-latitude (Petters and Wright, 2015). The dotted (M92), dashed (C86), dot-dashed (F62), and long-dashed (S21) lines indicate the INP parameterization from Meyers et al. (1992), Cooper (1986), Fletcher et al. (1962), and Schneider et al. (2021) respectively. The S21 (autumn) and S21 (spring) for Schneider et al. (2021) parameterization represent INP concentrations predicted during our autumn 2019 and spring 2020 campaign, given the average ambient temperature of -4.1 and -13.1 °C, respectively.

In addition, the overselling of our parameterization was tuned down by limiting its applicability (e.g., see revised manuscript lines 314-316) according to the referee's feedback. We address this issue corresponding to the referee's specific comments below.

**General comments:**

**First a general comment on having an appendix: The way the text is structured, it is necessary to go back and forth in the file to get all information during reading. Also, the appendix is quite long. I suggest to prepare supplemental information as a separate file. If I refer to the appendix below, this also always alternatively stands for supplemental information.**

We agree with the reviewer and have now lengthened the manuscript to include some of the previous appendix sections in the main text and have prepared an additional supplemental information file.

**Line 8-9: It would be informative to already know from the abstract, during which months you measured.**

Thanks. The related text (see lines 9, revised manuscript) is changed to *"...12 weeks of field measurements during October and November in 2019 and March and April in 2020"*

**Line 53: The method to fit INP concentration distributions at one temperature with a frequency distribution was first suggested by Welti et al. (2018), who was, to my knowledge, the first in the INP community, who referred to the paper by Ott et al. (1990), which you cite here. You cite Welti et al. (2018) later, so you should know the content of this work. I was waiting for Welti et al. (2018) to be cited already above, where you describe different types of fitting. Refer to Welti et al. (2020) and the fact that he fitted frequency distributions to INP data either above or related to this sentence here.**

We agree with the reviewer and have now cited Welti et al. (2018) in the same sentence as Ott (1990) (see line 56ff in the revised manuscript): *"Welti et al. (2018) observed INP concentrations follow the log-normal frequency distribution at investigated temperatures, explained by the successive random dilution model (Ott, 1990)."*

**Line 76-77: "A flow splitter (custom-built), a 0.5 m long 90°-bend and a three-way ball valve (...) connected a blower (...) and a high flow-rate impinger (...), both operating at 300 L min−1, downstream to the inlet." Please revise this sentence, as it is difficult to understand. Imagining the equipment, I guess I know what you mean, but was the bend in one of the lines after the flow splitter, or before? What was the diameter of the tubing? Maybe an additional sketch would be good as well.**

We thank the referee for pointing out the need for more information and adapted lines 80-86 in the initial manuscript to:

*Downstream of the inlet, a flow splitter (custom-built) directs the aerosol inflow into INP and aerosol instruments. For INP measurements, as shown in Figure 1b, the first sampling branch was regulated with a steady total flow of 300 L min⁻¹ through the inlet to the high flow-rate impinger (Coriolis® μ, Bertin Instruments, France). The inlet diameter was 50 mm which was tapered to 25 mm (KF-25 pipe standard) after the flow splitter. A detailed sketch of the applied setup has previously been used in Wieder et al. (2022b) Figure 1e. Secondary sampling lines that branched off the flow splitter had a diameter of 6 mm and were used by different instruments operating at flow rates between 0.283 std L min⁻¹ and 1 L min⁻¹. The INP sampling and auxiliary aerosol measurements are explained in more detail below.*

**Line 89-90: It remains totally unclear, how INP measurements were done. Liquid was sampled by the impinger, and then? I assume measurements were done right after 1 h sample collection? Or were the samples stored frozen? The measurements you did with DRINCZ, and basics of how DRINCZ works need to be explained at least shortly. This location here is the one where this information fits best.**

We agree with the reviewer and make the following changes on line 87ff.

*INPs were monitored using an offline method and an online method as follows. Using the Coriolis impinger with the cut-off size of 0.5 μm (aerodynamic diameter), ambient aerosol samples were collected into pure water. To compensate the evaporation loss of the sampling liquid (W4502-1L, Sigma-Aldrich, US) during the operation of the impinger, additional sampling liquid was fed into the sampling cone at a constant feed rate, which varied according to the ambient conditions and ranged between 0.6 and 1.0 mL min⁻¹. Between October 6 and November 15 in 2019, and from March 16 to April 22 in 2020, 137 and 133 samples, respectively, were collected and analyzed for INP concentration. The INP analysis was performed on site immediately after sample collection using the DRoplet Ice Nuclei Counter Zurich (DRINCZ, David et al., 2019). From each collected sample, 96 aliquots of 50 μL were pipetted into PCR trays and cooled down in the ethanol bath of a thermostat*

*(Figure 1b). During cooling, a camera mounted above the bath took pictures of the tray and the corresponding temperature of the bath was recorded. From the optical intensity difference of an aliquot between the two pictures, its freezing (temperature) was derived. From the impinger flow rate and aliquot volumes, an INP concentration can be derived. For further details we refer the reader to David et al. (2019).*

**I was also wondering, in Fig. 1 (b), where DRINCZ really is – I assume it is the thermostat. If so, move the word "DRINCZ" to the left, and also move "Coriolis impinger" to where this really is.**
Thanks. The position of the text for the two instruments is changed as requested by the reviewer.

**Line 90: The parameter N_INP is not officially introduced, therefore, behind "INP concentration" add "(N_INP)".**
Thanks. ($N_{INP}$) is added after "INP concentration" (see line 99 in revised manuscript).

**Equ. 1: You explain how to derive INP concentrations from impinger samples/DRINCZ measurements. But how about data from HINC?**
Thanks. We calculate the limit of detection from background counts (with filtered air) and use data only above the limit of detection to develop the parameterization. This is indicated in the manuscript (see lines 129-136 revised version).

In the original line 109 (now line 129ff in revised manuscript), we change the sentence *"...to determine background counts, which were used to determine the limit of detection (LOD) of the instrument at the measuring conditions (details in Lacher et al., 2017)."* to *"to determine the background ice particle concentrations. The INP concentrations are further derived by subtracting the background interference from sample measurements. Moreover, the limit of detection (LOD) of the instrument was also determined from the concentrations and standard deviations of the background interference measurements following Poisson statistics (detailed description in Lacher et al., 2017)."*

**Line 99: You refer to a background correction. Collecting samples in an impinger (with added water to keep its performance) will enrich impurities in the water in the samples. For a background correction, you likely collected particle free air into pure water in the impinger for the typical sampling time of 1 hour? Or did you determine this otherwise?**
As suggested by the reviewer, we determined this otherwise, see below

**In any case, it is needed to explain how you determined the background. Also add at least one plot (best in the appendix) with raw measured INP data (frozen fractions) from samples and from background, as suggested as good practice by Polen et al. (2018).**
We agree with the reviewer. To our understanding, there are two sources of potential contamination increasing the water background: (i) contamination on the surface of the sampling cone and (ii) contamination in the water and the tubing from the refilling system. To account for the two sources of contamination, a sampling cone was installed in the impinger and filled with water (15mL) entirely primed from the refilling system. Afterward, the cone was removed, sealed with a cap, and manually shaken for a minute and this water sample was analyzed for freezing in the same manner as the samples. We added Figure S3 in the Supplementary Information to show freezing curves from our background and samples and extended our discussion in line 109ff accordingly:

[Figure]

**Figure S3.** Overview of frozen fractions as a function of temperature obtained with DRINCZ (David et al., 2019) of background samples (blue) and INP samples (orange).

*[...] is the sampling time (60 minutes), $V_{Coriolis}$ volume within the sampling cone (15 mL), $p_{ambient}$ and $p_{std}$ (1013.25 hPa) are the ambient and standard-condition pressure, and $T_{ambient}$ and $T_{std}$ (273.15 K) are the ambient and standard-condition temperature. The retrieved INP concentrations were corrected for the background of blank samples according to David et al. (2019) as follows. Blank samples were taken by installing an unused sampling cone in the impinger and filling it with water (15 mL) entirely primed from the refilling system to account for contamination originating from the refilling system and any impurities in the water. The cone was removed, capped, and manually shaken for a minute to account for contamination from the cone surface. A blank sample was taken every three days during the campaigns. For each season, a fit of the backgrounds of the season was used to correct the INP samples of the corresponding season. Following Vali (2019), INP samples were background corrected by subtracting the differential INP spectrum of the blank fit from an INP sample's differential INP spectrum. An overview of the raw frozen fractions of INP samples and background blanks as input for the calculation of the differential INP spectrum are presented in Figure S3. From the analysis by DRINCZ, the highest temperature at which the INP concentration could be measured was approximately -5 °C, while the coldest temperature where ice nucleation activity was reliably observed was -22 °C.*

**Line 104: Did HINC use a pre-impactor to reduce false signals from large aerosol particles, as it is typically done for these in-situ instruments? Please state this in the text.**

We did not install a pre-impactor upstream of HINC because the upper size cut-off and the OPC size channel used to classify ice should preclude large particles from being classified as ice. To make this clearer, in line 124 in the revised manuscript we added *"The ice detection threshold of 5 µm is unaffected by inactivated large ambient particles given HINC's upper cut-off size of approximately 2.5 µm ($D_{50}$, i.e., 50 % loss for particles with a diameter of 2.5 µm) due to the horizontal orientation of the chamber."*

**Line 123: This whole chapter 2.2 is difficult to follow and has a quite technical content. It may be better to add a more describing chapter here, in which you roughly explain what you did, and move this rather technical chapter to the appendix. But no matter if you move most of the following content to the appendix or not, the text here needs to start with one or a few explanatory sentence(s) about what will be introduced in the following.**

We agree with the reviewer. The original content in Chapter 2.2 is moved to the supplementary information S1. Instead, this chapter is rewritten to be more descriptive in line 146ff: *"In this study, we aimed to predict the INP concentrations solely as a function of the observed nucleation temperatures (explained in Section 3.2). Using linear regression to fit the relationship between the logarithmic space of INP concentrations and nucleation temperature requires the ordinary least square technique to estimate the regression coefficients. A critical assumption behind this method is homoscedasticity, i.e., constant variance of errors for all observations regardless of the value of regressors. However, in reality, this is hardly achieved, in particular for real-time observational data in the field, since INP concentrations naturally vary several orders of magnitude at a given temperature. In addition, the INP concentrations at each measured temperature are not evenly distributed, which are limited by inter-instrumental differences and available sample sizes. As a*

*result, we observed a non-constant variance in the errors (i.e., heteroscedasticity) of INP concentrations over the investigated temperatures (See Fig. S1 in the Supplementary Information). We, therefore, applied weighted factors in fitting our parameterization to obtain unbiased regression coefficients. The details are given in Section S1 of the Supplementary Information."*

**Line 168: "indistinguishable" may be the wrong word, here. Also, at the end of the sentence add "for the months during which we did our measurements." This is important (as elaborated on repeatedly in this review), as there is a seasonal variation in the Arctic (also observed at Ny-Alesund, see Wex et al., 2019), and that is exactly one of the reasons why your parameterization may not be as broadly applicable as you suggest in your text, performing worse when comparing to literature data taken in winter and summer.**
We agree. The text is changed to *"Overall seasonal variation of INP concentrations at the investigated temperatures is unidentifiable for the months during which we sampled INP concentrations."*

**Figure 4: Do I get it right, that D15 was shown as solid line only in the "applicable temperature range" at -20°C and -30°C, and else is given at dotted line? Then add the dotted line with a description to the legend. (The same holds for Fig. A3.) Also, maybe use brighter blue and green colors for the text in the plots (concerning rho and p). (This also applies for other respective plots.)**
We thank the referee for pointing out the need for more clarity. The referee is correct that solid lines represent parameterizations used within their applicable temperature ranges and dashed lines if not. Based on the comments above, we changed Figure 4 and adapted the parameterization labeling to it. Note that the information on the applicable temperature range is still given in the caption. We agree that more pronounced colors improve the readability and changed them consistently throughout the manuscript.

**Line 190: "For a few decades" – if this is true, then why do you cite two papers from 2021 on this issue? And what do you mean by "temperature dependence"? It seems to imply that knowing the slope of the curve is correct. But I guess that is not what is meant? Revise the test to be more specific on that.**
Thanks. The references (line 228 in the revised manuscript) are changed to *"(Fletcher et al., 1962; Cooper, 1986; Meyers et al., 1992)"*. To clarify the temperature dependence of INP concentrations, the sentence is changed to *"We present a methodology to optimally fit the slope of INP concentrations as a function of investigated nucleation temperature from their frequency distributions."* (lines 228-229 in the revised manuscript).

**Line 203: "To our knowledge, it is the first attempt to predict INP concentrations in the Arctic utilizing in-situ measurements." Collecting a sample for 1 hour and then examining the resulting bulk liquid (which is, I think, you did with DRINCZ) is not an in-situ measurement. Also, Schrod et al. (2020a), which you cite a number of times, gave an INP concentration parameterization, albeit based on relative humidity. Nevertheless, your statement is incorrect. Revise this sentence or remove it completely.**
We agree with the reviewer that the statement is not accurate, therefore has been removed in the revised manuscript.

**Line 206: Here you compare to the Fletcher, Cooper and Meyers parameterizations and state that your parameterization is lower than these. It may be good to add that these older parameterizations were done for different environments (i.e., not the transition months in the Arctic).**
We agree with the reviewer. The sentence is added in line 206 (updated version lines 249-250): *"Note that these three parameterizations were derived from the observations in different environments (not during the seasonal transition months in the Arctic)."*

**Figure 7: This relates to my main concern #3. Either include SPIN data from Hartmann et al. (2021) in both figure and analysis or give a good reason for omitting it.**
We agree with the reviewer. As mentioned above, in the response to the 3rd point for major issues, the SPIN data is now added to the evaluation and corresponding figures (Fig. 8 and 10) now.

**Line 216ff: This section about how well the parameterization fits, regardless of season and in general needs to be revised, also maybe in light of the addition of SPIN data, which may influence overall performance of your parameterization, or at least its performance during summer.**

We agree and the reviewer is right in pointing out that the inclusion of SPIN data deteriorates slightly the overall performance (percentage of predicted INP concentrations falling within the 95 % confidence interval from *81.0 %* to *80.2 %*), especially in summer (from *75.9 %* to *74.5 %*) since the INP concentrations measured by SPIN were systematically higher compared to our prediction ranges (Figure 10 in the revised manuscript). The sentence in line 216 *"Regardless of the season, location, measuring instrumentation, and the nature of variation of INP concentrations, the trend of the INP concentrations is well captured by our parameterization."* was removed, and the Hartmann 2021 data was also discussed more carefully by differentiating between the filter and SPIN data. In addition, a sentence was added in line 300 in the revised manuscript regarding SPIN data: *"However, our parameterization underestimates the INP concentrations measured in SPIN's temperature range (Hartmann et al., 2021} during summer, which could be explained by the increased local terrestrial source (e.g., mineral dust) in the season when the surface is free of ice and snow."*

**Line 233ff: A possible regional dependence of your results, which you suggest here, could be of interest. From your big data-set, did you try to compare terrestrially collected data with other data (from ships or airborne)?**

We agree with the reviewer. We added Figure 9 in the revised manuscript summarizing the INP concentrations at -15 °C by classifying the Arctic campaign data based on measuring platforms. In addition, a descriptive paragraph was added to Section 3.4 line 269ff:

*"In addition to the seasonal differences, we summarized the INP concentrations at -15 °C by classifying the Arctic INP concentration data by measurement platform (Figure 9). The measured INP concentration ranges overlap between different measuring platforms, except for Prenni et al., 2009 who observed systematically higher INP concentrations (air-borne) compared to other studies, possibly due to the presence of mineral or soil dusts. The overall level of ground-based measured INP concentrations was slightly higher than the ship-borne measurements at this temperature, indicating a relative enhancement from terrestrial INP sources. Besides, only two studies (Prenni et al., 2009 and Hartmann et al., 2020) measured INP concentrations from flight campaigns, challenging the comparison with the other two categories (i.e., ground-based and ship-borne)."*

In addition, Figure R1 was added (Fig. S7 in the supplemental information file in the revised manuscript) to compare the predicted concentrations to previous observations classified by the platforms the measurements were conducted. From the percentage of predicted data falling within the 95 % confidence intervals, a notable trend of the predicted INP concentrations on any of the measuring platforms was not observed.

[Figure]

Figure 9. Comparison of INP concentrations at -15 °C from this study to selected Arctic field campaigns as a function of measurement platform. The median $N_{INP}$ is given in colored dots, and the colored error bars indicate the 5 – 95 % quantile of the corresponding data set.

[Figure]

Figure R1 (S7). Predicted INP concentrations from the proposed fit (Equation 3) compared to observations from previous Arctic field campaigns per measuring platform (i.e., ground-based, ship-borne and air-borne). The values in the parenthesis represent the percentages of predicted data falling within the 95 % confidence intervals.

**Line 251: State clearly, following "95% confidence interval", that this confidence interval covers two orders of magnitude.**
The sentence (see line 313 in the revised manuscript) is changed to: *"...within 95 % confidence interval that covers approximately two orders of magnitude…"*

**Line 254-256: What do you mean by well-mixed INP air masses? Also: the reasoning of the sentence here is not logical. INP are so rare, that most "overall" aerosol properties will not be useful to parameterize them. This means that a good aerosol parameter which can predict INP has not yet been found (and we may not find one), but it says nothing about the mixing state of air masses wrt. INP. At most, it may indicate that the majority of aerosol particles and the majority of INP may have different sources. This sentence needs a thorough revision.**
We agree with the reviewer that this statement was not clear. What we meant by well-mixed was the absence of a specific source of aerosol or an airmass dominated by a particular aerosol species. We have now rephrased this part to (see lines 314-316 revised manuscript)

*"Note that the presented INP parameterization is specified for the Arctic environment particularly relevant for the autumn and spring transition seasons where no particular aerosol type dominates the INP concentrations."*

**Line 259-261: Given the limitations I listed in this review, this sentence needs to be tuned down. It sounds like you are trying to sell something. Particularly, I don't see how you can make any inference about a vertical INP profile!**
We agree that this sentence is poorly phrased. What we meant is that our parameterization can be used to predict INP concentration as a function of temperature in the troposphere. We have revised the sentence to (see line 321-324 revised manuscript):

*"We hope our INP parameterization promotes future modeling studies via a more simplistic prediction of INP concentrations in the Arctic environment as a function of temperature, particularly during the transition seasons of fall and spring thus improving the representation of MPCs and Arctic climate"*

**Line 256-257: "new INP parameterization can be used as a proxy to estimate the pre- industrial or pristine INP level" This was not discussed in the text at all, and is highly questionable. Remove this text. Besides**

for this not being a topic in your study at all, here some more reasoning: You do not know how "pristine" your data is, given that it was collected on land, and then close to a settlement in the Arctic (Ny-Alesund). As for pre- industrial times, while it has been argued that Arctic haze does not add INP to Arctic air masses (at least in the temperature range which also you are looking at), it is not clear, to my knowledge, if overall INP concentrations have or have not changed. I know of two studies examining Arctic ice cores (Hartmann et al., 2019 and Schrod et al., 2020b – not intended for inclusion in your work, I only need them for my argument here), which however only examined ice cores dating up to ~ 1990, which is before Arctic amplification clearly started to show. These two studies also come to differing conclusions, the former saying that up to this time, no change was observed, while the latter said that in the modern-day period for temperatures < -22°C, there were higher and more variable INP concentrations. So overall, nothing is known about the performance of your parameterization to describe pre-industrial times.**

We agree with the reviewer's comment and reasoning above and have removed the following text from the revised manuscript "*can be used as a proxy to estimate the pre-industrial or pristine INP level, and it*"

**Fig. A1: This figure isn't referred to in the appendix, only in the main text, which I find confusing. At the same time, I found it confusing that the appendix starts with referencing to Fig. A2. I suggest to add a paragraph on the fitting as A1, which would bring all that in order.**

We agree, by moving the details of the parameterization approach (Previous Section 2.2) to the Supplementary Information, Figure S1 is referenced prior to Figure S2 now with descriptions in the updated Section S1 in the Supplementary Information.

**Editorial comments:**

**Line 115: Mention in section 2.1.2 explicitily, that the WIBS instrument was only used during fall. Right now, one has to get down to A2 to know about that, which is too late.**

We agree and now revise it as follows. "*Note that WIBS measurements were only available during the autumn campaign in 2019.*" is added to the end of Section 2.1.2 now (lines 144-145 in the revised manuscript).

**Line 130: Delete "the" before "heteroscedastic".**

Thanks, and done (see line 8 in the Supplementary Information).

**Line 158-159: "The overview of the measurement site and the experimental setup are given in the Section 2.1 and Fig. 1a." - This doesn't need to be repeated here and is rather interrupting the flow of the text -> remove this sentence.**

Thanks. This sentence is removed (see line 163 in the revised manuscript).

**Line 161: In the parenthesis, "further" should not be capitalized.**

Thanks. It is changed to a small case now (see line 164 in the revised manuscript).

**Line 165: What do you mean by "virtually log-linear pattern"? Maybe delete "virtually"??? Line 165: Delete "the" before "decreasing".**

Thanks. Changes are made accordingly (see line 168 in the revised manuscript).

**Caption of Fig. 4: "as a function of the particle larger than 0.5 μm (volume equivalent diameter) number concentration (n>0.5μm)" should better be something like "as a function of n>0.5um (the number concentration of particles larger than 0.5um volume equivalent diameter)".**

Thanks. Changes are made accordingly (see updated caption of Fig. 4).

**Line 166-167: "we conducted unpaired t-tests with a significance level of 5 % for observed INP concentrations at each temperature" I know what you did, here, but maybe this could be formulated a bit better.**

Thanks. We reformulate the sentence (see lines 169-170) to "*For INP concentrations observed in autumn 2019 and spring 2020, we performed unpaired t-tests to infer if there is a significant seasonal difference at the 95% confidence interval level (p < 0.05).*"

**Line 186 ff: "relations were first fitted between ... relations were further linked with temperatures" – this is not the correct description. Typically, a parameter like the surface site density n_s was calculated based on measured (temperature dependent) INP concentrations together with the surface area, and then the obtained (temperature dependent) calculated values were fitted- so these two sentences here are off and need rewording.**

We thank the referee for pointing out the inaccuracies that were introduced in this part. We rephrase (see line 224ff) the passage to: "*The improved accuracy of advanced INP parameterizations relies on a robust relationship between INP concentration and aerosol or meteorological properties – which was not evident in our in-situ observations (Section 3.2). Alternatively, average INP concentrations can be predicted solely by a nucleation temperature dependent parameterization.*"

**Line 189: Add "area" after "surface".**

Done, and all instances of surface have now been replaced with "*surface area*"

**Line 191: First word should be in plural: "parameterizations".**

Thanks. Changes are made accordingly (see line 227 in the revised manuscript).

**Line 191-192: "We present a methodology to optimally fit the temperature dependence of INP concentration from frequency distributions." Sounds somewhat strange, think about rewording. Part of my bad feeling about this may be that you don't fit a temperature dependence, but the frequency distributions.**

Thanks. To clarify the temperature dependence of INP concentrations, the sentence in line 191 is changed to "*We present a methodology to optimally fit the slope of INP concentration frequency distributions as a function of investigated nucleation temperature.*" (Lines 228-229 in the revised manuscript)

**Line 198: Delete "the" before "trimmed".**

Thanks. Change is made accordingly in line 54 in the revised manuscript in the supplementary information.

**Line 206: Add "than these parameterizations" at the end of the sentence.**

Thanks. Changes are made accordingly in line 248 in the revised manuscript.

**Line 207: Replace "overestimated" with ""overestimating our values".**

Thanks. Changes are made accordingly in line 251 in the revised manuscript.

**Caption of Fig. 5: "inidcated" needs to be corrected. AND add an "are" after "parameterizations".**

Thanks. Changes are made accordingly in Figure 6 caption in the revised manuscript.

**Figure 5 versus Figure 7: Check which line is which, and which ones are correct – Fletcher, Cooper, and Meyers lines should be the same in these two plots, but they are different, at a first glance. (It could be that Fletcher is different?)**

Thanks for noticing this. Despite different scales, the Fletcher 1962 parameterization in previous Figure 5 was not correctly plotted. We made corrections as Figure 6 in the revised manuscript correspondingly.

**Figure 7: In the legend, Ny-A..lesund (at Schrod 2020) needs to be corrected.**

Thanks. It is corrected in Figure 8 in the revised manuscript.

**Figure 7: "The temperature in parenthesis of the Schneider et al. (2021) parameterization represent the average ambient temperature observed during the autumn (-4.1 °C) and spring (-13.1 °C) campaigns." - I don't see any temperature in parenthesis for the Schneider et al. (2021) parameterizations. Please check the figure or change this sentence.**

Thanks. The corresponding caption text in Figure 7 (Figure 8 in the revised manuscript) has changed to "*The S21 (autumn) and S21 (spring) for Schneider et al. (2021) parameterization represent INP concentrations*

*predicted during our autumn 2019 and spring 2020 campaign, given the average ambient temperature of - 4.1 and -13.1 °C, respectively."*

**Line 238: To describe SPIN, change the text in the parenthesis with "(an online instrument based on the same measurement principle as HINC)".**
Thanks. Changes are made accordingly (see line 296 in the revised manuscript).

**Line 249: In the text you typically use the expression of a 95% confidence interval, here you now refer to 2sigma. Might be better to have this consistent throughout the text.**
Thanks. In the parenthesis "*2σ*" is changed to "*95 % confidence interval*" for consistency (see line 309).

**Line 271: You did not measure the surface (that could be done with an epiphaniometer), but calculated it. Therefore, delete "and surface", or mention that you derived it.**
Thanks. "*and surface*" is removed from the original text in line 43 in the revised manuscript in the supplementary information.

**Line 271: Sentence starting with "Generally" misses a verb ("was observed" or so).**
Thanks. "*was observed*" is added after "*a weak correlation*" in line 44 in the revised manuscript in the supplementary information.

**Line 274: "concentration of fluorescent particle concentration" - delete one "concentration".**
Thanks. Changes are made accordingly in line 46 in the revised manuscript in the supplementary information.

**Page 15, caption for Tab. A2: As the abbreviation was only used here, give the explanation "confidence interval" for "CI".**
Thanks. Explanations were added accordingly (see Table 1 in the revised manuscript).

**Figure A5: Concerning this figure, it is only mentioned in the text that it exists. Add a paragraph or so (in the appendix), at least explaining what it shows.**
We agree with the reviewer. A paragraph with descriptions of Figure A5 (now Figure S6 in the revised manuscript) is added in Section S5 in the Supplementary Information (line 49ff in the revised manuscript in the supplementary information): "*A QQ (quantile-quantile) plot provides a statistical solution to verify and visualize the hypothesized distribution (log-normal distribution in this study) of given random variables. For the variables in this study, i.e., INP concentrations at every measured temperature, the integral percentiles (1 %, 2 %,..., 100 %) of the observed distribution were computed. They were consequently plotted against those from a theoretical log-normal distribution (see Fig S6). The closer the scatter points lie in a straight line, the more identical the type of distribution to the theoretical distribution for the selected variables. In Figure S6, the log-normal distribution is more evident for cold nucleation temperatures, especially when T < -12 °C. Nevertheless, the trimmed tails of the distributions can be identified at higher temperatures T ≥ -12 °C), where INP concentrations are biased towards the minimum detectable concentration.*"

**Literature:**

Creamean, J. M., R. M. Kirpes, K. A. Pratt, N. J. Spada, M. Maahn, G. de Boer, R. C. Schnell, and S. China (2018), Marine and terrestrial influences on ice nucleating particles during continuous springtime measurements in an Arctic oilfield location, Atmos. Chem. Phys., 18, 18023–18042, doi:10.5194/acp-18-18023-2018.

Hartmann, M., T. Blunier, S. O. Brügger, J. Schmale, M. Schwikowski, A. Vogel, H. Wex, and F. Stratmann (2019), Variation of ice nucleating particles in the European Arctic over the last centuries, Geophys. Res. Lett., 46, doi:10.1029/2019GL082311.

Hartmann, M., X. Gong, S. Kecorius, M. van Pinxteren, T. Vogl, A. Welti, H. Wex, S. Zeppenfeld, H. Herrmann, A. Wiedensohler, and F. Stratmann (2021), Terrestrial or marine? – Indications towards the origin of Ice Nucleating Particles during melt season in the European Arctic up to 83.7°N, Atmos. Chem. Phys., 21, 11613-11636, doi:10.5194/acp-21-11613-2021.

Ott, W. (1990), A physical explanation of the lognormality of pollutant concentrations, J. Air Waste Manag. Assoc., 40(10), 1378-1383, doi:10.1080/10473289.1990.10466789.

Polen, M., T. Brubaker, J. Somers, and R. C. Sullivan (2018), Cleaning up our water: reducing interferences from nonhomogeneous freezing of "pure" water in droplet freezing assays of ice-nucleating particles, Atmos. Meas. Tech., 11, 5315–5334, doi:10.5194/amt-11-5315-2018.

Porter, G. C. E., M. P. Adams, I. M. Brooks, L. Ickes, L. Karlsson, C. Leck, M. E. Salter, J. Schmale, K. Siegel, S. N. F. Sikora, M. D. Tarn, J. Vüllers, H. Wernli, P. Zieger, J. Zinke, and B. J. Murray (2021), Highly active ice-nucleating particles at the summer North Pole, ESSOAr, doi:https://www.essoar.org/doi/10.1002/essoar.10508073.1.

Sanchez-Marroquin, A., O. Arnalds, K. J. Baustian-Dorsi, J. Browse, P. Dagsson- Waldhauserova, A. D. Harrison, E. C. Maters, K. J. Pringle, J. Vergara-Temprado, I. T. Burke, J. B. McQuaid, K. S. Carslaw, and B. J. Murray (2020), Iceland is an episodic source of atmospheric ice-nucleating particles relevant for mixed-phase clouds, Science Advances, 6(26), doi:10.1126/sciadv.aba8137.

Schrod, J., E. S. Thomson, D. Weber, J. Kossmann, C. Pohlker, J. Saturno, F. Ditas, P. Artaxo, V. Clouard, J. M. Saurel, M. Ebert, J. Curtius, and H. G. Bingemer (2020a), Long-term deposition and condensation ice-nucleating particle measurements from four stations across the globe, Atmos. Chem. Phys., 20(24), 15983-16006, doi:10.5194/acp-20-15983-2020.

Schrod, J., D. Kleinhenz, M. Horhold, T. Erhardt, S. Richter, F. Wilhelms, H. Fischer, M. Ebert, B. Twarloh, D. Della Lunga, C. M. Jensen, J. Curtius, and H. G. Bingemer (2020b), Ice-nucleating particle concentrations of the past: insights from a 600-year-old Greenland ice core, Atmos. Chem. Phys., 20(21), 12459-12482, doi:10.5194/acp-20-12459-2020.

Tobo, Y., K. Adachi, P. J. DeMott, T. C. J. Hill, D. S. Hamilton, N. M. Mahowald, N. Nagatsuka, S. Ohata, J. Uetake, Y. Kondo, and M. Koike (2019), Glacially sourced dust as a potentially significant source of ice nucleating particles, Nat. Geosci., 12(4), 253-+, doi:10.1038/s41561-019-0314-x.

Welti, A., K. Müller, Z. L. Fleming, and F. Stratmann (2018), Concentration and variability of ice nuclei in the subtropical maritime boundary layer, Atmos. Chem. Phys., 18, doi:10.5194/acp-18-5307-2018.

Wex, H., L. Huang, W. Zhang, H. Hung, R. Traversi, S. Becagli, R. J. Sheesley, C. E. Moffett, T. E. Barrett, R. Bossi, H. Skov, A. Hünerbein, J. Lubitz, M. Löffler, O. Linke, M. Hartmann, P. Herenz, and F. Stratmann (2019), Annual variability of ice nucleating particle concentrations at different Arctic locations, Atmos. Chem. Phys., 19, 5293–5311, doi:10.5194/acp-19-5293-2019.

---

## Author Comment (AC2)

**Referee comments 2**

We thank referee 2 for the valuable feedback on our manuscript acp-2022-21. In response to the questions and suggestions, please find our answers and corrections listed below. **referee's comments are reproduced in bold** and author responses in the normal font; *extracts from the original manuscript are presented in red italic*, and *from the revised manuscript in blue italic*.

**Review of "Predicting atmospheric background number concentrations..." by Li et al.**

**The authors report INP measurements at Ny-Alesund (Svalbard) at temperatures between 0 and -30 °C for 12 weeks (Oct-Nov and March-April). They did not see a significant difference in INP concentrations between Oct-Nov and March-April. In addition, the results fall within the range of INP concentrations previously reported for the Arctic. Also, they show that parameterizations developed for mineral dust, sea salt aerosol, and bioaerosols over pine forests do not predict the measurements well. Also, other parameterizations developed from measurements in other regions do not describe the measurements well. In addition, they fit their data as a function of temperature and suggest that this parameterization could be used to describe background number concentrations in the Arctic.**

**The measurements at Svalbard are important, and I congratulate the authors for generating an important dataset. I think it is also useful to show that these measurements cannot be reproduced with previous parameterizations, even though the disagreement is expected since the parameterizations were developed for other regions or for specific types of aerosols. I support the publication of this part of the manuscript, although publication as a Measurement Report may be more appropriate than publication as a Research Article.**

We thank the reviewer for their positive evaluation of our manuscript.

**Like Referee #1, I have major concerns about the section that describes the new parameterization of the authors' data. The authors are suggesting that their measurements from one location in the Arctic and during only 12 weeks can be used to predict background concentrations for the entire Arctic. To me, it does not make a lot of sense to use measurements from one location in the Arctic and during only 12 weeks to develop a parameterization for the entire Arctic, especially when some studies have shown a seasonal dependence on INP. If one wanted to make a parameterization for the entire Arctic, it would make more sense to use all the currently available INP data collected in the Arctic. I am not suggesting doing this, since INP concentrations are expected to change with season and location in the Arctic (at least in some cases), and one parameterization using all the previous measurements would miss this variability.**

We agree with Reviewer 2 in this regard and have toned down the statements regarding the general applicability of our parameterization for the Arctic. Rather, we specify that our data fills in a gap in the literature for INP concentrations during the transition seasons (fall and spring) where we believe our data and parameterization are most useful. This point also explains the absence of seasonality observed in our data set compared to those reported in the literature, which are mostly comparing summer to winter seasons.

**Below are specific comments:**

**Figure 7 shows that the parameterization developed by the authors is off by approximately 2 orders of magnitude in some cases. To claim the new parameterization is doing a good job for the entire Arctic seems to overstate the power of the new parameterization. Furthermore, if you fit all the previous Arctic data, you would get a different parameterization and one that would be likely more applicable to the whole Arctic, since it would be based on measurements for different locations and many different times of the year. Why develop a parameterization from just one location and for a very limited time?**

We agree with the reviewer here that to develop a parameterization valid for all Arctic regions and all seasons, it would make sense to use all available data. To tackle this, we now specify in the conclusions section (see lines 314-316 in the revised manuscript) and the abstract (lines 10-15 in the revised manuscript) of the revised manuscript that our parameterization is useful for the transition seasons and explain that this could be one of the reasons why we do not observe a seasonal effect in INP concentrations. Furthermore, to justify the use of the simplistic T-dependent parameterization as we have presented in our paper, we would also require showing that all other INP concentrations from the literature in the Arctic are also not well predicted by their respective aerosol and meteorological parameters. However, this would be beyond the scope of the current paper, as it would require also collecting the raw high time resolution aerosol and meteorological data from all published studies.

**Page 1, line 10. Consider changing "not feasible" to "not successful".**
We agree with the reviewer and have changed the wording accordingly (See line 12 in the revised version).

**Page 4, lines 61-63. "This parameterization will help evaluate the role of cloud phase interactions in Arctic MPCs, and contribute to the progress on accurately estimating cloud influenced climate predictions in the Arctic". Like Referee #1, I think the authors are overselling their work here. They have done measurements at only one location for 12 weeks, and then they used this data to develop a parameterization for the whole Arctic. A more reasonable approach would be to take all the previous measurements in the Arctic and then generate a parameterization based on this combined data set. Even then, I do not think this is a very useful approach since previous work has shown that concentrations can vary with season and location in the Arctic (at least in some cases).**

We agree with the reviewer and refer them to the response above in part. We agree that higher INP concentrations were observed previously, especially in summer when the surface is free of ice and snow cover, thus pronounced local sources (e.g., dust, biological INPs) were expected. Despite the importance due to augmented local sources in summer, only a few Arctic INP measurement campaigns (less than 9 % of total reference data) focused on transition seasons, highlighting the need for more measurements during the transition season. In this context, seasonal variation was not observed from our measurements. We have now acknowledged the limitation of our parameterization in the paper (see responses above). In addition, we note that the distribution-based parameterization provides useful information on a general level and range of INP concentrations in the Arctic environment, which could be applied to regional climate models for long-term predictions. In this regard, several sentences are added to discuss the discrepancy in line 216 (revised manuscript line 261ff): "*However, a few studies (e.g., Wex et al., 2019, Tobo et al., 2019, Creamean et al., 2018) observed consistently higher INP concentrations in the summer, indicating enhanced sources of local emissions due to the decreased ice cover. We note that the parameterization herein was derived from the measurements during transition seasons (autumn and spring), aiming to predict the general level and range of INP concentrations in the Arctic. Therefore, applying it to generate INP concentrations in specific seasons, particularly the Arctic summer, could introduce a low bias.*"

**Page 3. Line 84. The Coriolis impinge has a cut-off of 0.5 μm. Does this mean the technique is missing a large fraction of INPs? Could this explain the lack of agreement with the parameterizations based on particles above 0.5 μm? Please discuss. Did the HINC have a similar cut-off?**

The cut-off size of 0.5 μm represents the lower bound for the Coriolis impinger, i.e., the impinger efficiently collects particles with aerodynamic diameters larger than 0.5 μm. "*(lower limit)*" is added in the text for clarification. Therefore, it is adequate to link the INP parameterization to the particles above 0.5 μm. So, this would not explain why there is a lack of agreement with other parameterizations based on particles above 0.5 μm (see Fig. S2 in the supplementary information). However, HINC has a different cut-off with an upper bound $D_{50}$ of approximately 2.5 μm (Lacher et al., 2017), i.e., it efficiently samples particles smaller than 2.5 μm. In this regard, we note that the concentration of particles above 2.5 μm is negligible.

**Page 13. Line 259-261. "Our INP parameterization promotes future modeling studies via a more realistic microphysical representation in the Arctic MPCs, especially the vertical profile of primary ice distribution**

**(Hawker et al., 2021), thus, improving the predictions for the future Arctic climate." Again, similar to Referee #1, I think the authors are overselling their results here.**

We agree with the reviewer and have modified this sentence completely to see lines 321-324 in the revised manuscript. The new sentence reads "*We hope our INP parameterization promotes future modeling studies via a more simplistic prediction of INP concentrations in the Arctic environment as a function of temperature, particularly during the transition seasons of fall and spring thus improving the representation of MPCs and Arctic climate.*"

**Page 13, lines 251-253. "Note that the presented INP parameterization is specified for the Arctic environment, where the atmosphere is well-mixed and transient effects average out." What is the evidence for a well-mixed atmosphere in the Arctic? I think there is a lot of field data that shows that the Arctic is not a well-mixed system. Please correct me if I am wrong.**

Thanks. We agree with the referee. We have modified the statements regarding the well-mixed to specify that the INP concentration is not dominated by a single aerosol species. See lines 314-316 in the revised manuscript.

**Page 13, lines 254-256. "The presence of well-mixed INP air masses is exhibited by the absence of a relationship with aerosol properties and further by the inability of previous aerosol-based INP parameterizations to reproduce the observations from this study." I do not understand the authors' logic here. The absence of a relationship with aerosol properties may have nothing to do with a well-mixed INP air mass. Furthermore, the inability of previous aerosol-based INP parameterizations to reproduce the observations from this study may have nothing to do with a well-mixed INP air mass. Please correct me if I am wrong.**

We agree with the reviewer here and similar to the response above, we have modified the text in the manuscript to reflect that the absence of INP dependence on aerosol parameters could be caused by the absence of a major dominating aerosol species in the air mass (see lines 316ff in the revised manuscript).